# Small-molecule activation of TFEB alleviates Niemann–Pick disease type C via promoting lysosomal exocytosis and biogenesis

Kaili Du[1,2†], Hongyu Chen[1†], Zhaonan Pan[1], Mengli Zhao[1], Shixue Cheng[1], Yu Luo[1], Wenhe Zhang[1], Dan Li[1,2]*

[1]Collaborative Innovation Center of Yangtze River Delta Region Green Pharmaceuticals, College of Pharmaceutical Sciences, Zhejiang University of Technology, Hangzhou, China; [2]Department of Molecular, Cellular, and Developmental Biology, University of Michigan, Ann Arbor, United States

## eLife Assessment

This study reports that activation of TFEB promotes lysosomal exocytosis and clearance of cholesterol from lysosomes, the strength of evidence for which is **convincing** with appropriate and validated methodology in line with current state-of-the-art. The significance of the findings is **important** in the context of Niemann–Pick disease type C as well as other subfields.

**\*For correspondence:**
lidan@zjut.edu.cn

[†]These authors contributed equally to this work

**Competing interest:** The authors declare that no competing interests exist.

**Abstract** Niemann–Pick disease type C (NPC) is a devastating lysosomal storage disease characterized by abnormal cholesterol accumulation in lysosomes. Currently, there is no treatment for NPC. Transcription factor EB (TFEB), a member of the microphthalmia transcription factors (MiTF), has emerged as a master regulator of lysosomal function and promoted the clearance of substrates stored in cells. However, it is not known whether TFEB plays a role in cholesterol clearance in NPC disease. Here, we show that transgenic overexpression of TFEB, but not TFE3 (another member of MiTF family) facilitates cholesterol clearance in various NPC1 cell models. Pharmacological activation of TFEB by sulforaphane (SFN), a previously identified natural small-molecule TFEB agonist by us, can dramatically ameliorate cholesterol accumulation in human and mouse NPC1 cell models. In NPC1 cells, SFN induces TFEB nuclear translocation via a ROS-Ca$^{2+}$-calcineurin-dependent but MTOR-independent pathway and upregulates the expression of TFEB-downstream genes, promoting lysosomal exocytosis and biogenesis. While genetic inhibition of TFEB abolishes the cholesterol clearance and exocytosis effect by SFN. In the NPC1 mouse model, SFN dephosphorylates/activates TFEB in the brain and exhibits potent efficacy of rescuing the loss of Purkinje cells and body weight. Hence, pharmacological upregulating lysosome machinery via targeting TFEB represents a promising approach to treat NPC and related lysosomal storage diseases, and provides the possibility of TFEB agonists, that is, SFN as potential NPC therapeutic candidates.

## Introduction

Lysosomes are essential organelles for the degradation and recycle of damaged complex substrates and organelles (*Xu and Ren, 2015*). In recent years, lysosomes have emerging roles in plasma membrane repair, external environmental sensing, autophagic cargo sensing, and proinflammatory response, thereby regulating fundamental processes such as cellular clearance and autophagy

(*Tsunemi et al., 2019*). Mutations in genes encoding lysosomal proteins could result in more than approximately 70 different lysosomal storage disorders (*Fraldi et al., 2016*). Niemann–Pick disease type C (NPC) is a rare lysosomal storage disorder caused by mutation in either *NPC1* or *NPC2* gene. Deficiency in NPC1 or NPC2 protein results in late endosomal/lysosomal accumulation of unesterified cholesterol (*Sarkar et al., 2013*; *Spampanato et al., 2013*). Clinical symptoms of NPC include hepatosplenomegaly, progressive neurodegeneration, and central nervous system dysfunction, that is, seizure, motor impairment, and decline of intellectual function (*Carstea et al., 1997*). So far there is no FDA-approved specific therapy for NPC, although miglustat, approved to treat type I Gaucher disease, has been used for NPC treatment in countries, including China, Canada, and the European Union (*Pineda et al., 2009*; *Pineda et al., 2010*; *Wraith et al., 2010*; *Chien et al., 2013*).

Transcription factor EB (TFEB), a member of the microphthalmia/TFE transcription factor (MiTF) family, is identified as a master regulator of lysosome and autophagy by controlling the 'coordinated lysosomal expression and regulation' (CLEAR) network, covering genes associated to lysosomal exocytosis and biogenesis, and autophagy (*Sardiello and Ballabio, 2009*; *Martini-Stoica et al., 2016*; *Napolitano and Ballabio, 2016*). In normal conditions, TFEB is phosphorylated by mTOR kinase and kept in cytosol inactively (*Martina and Puertollano, 2018*, *Napolitano et al., 2018*). Under stress conditions, that is, starvation or oxidative stress, TFEB is dephosphorylated and actively translocates into the nucleus, promoting the expression of genes associated with lysosome and autophagy (*Medina et al., 2015*; *Zhang et al., 2016*; *Puertollano et al., 2018*). TFEB overexpression or activation results in increased number of lysosomes, autophagy flux, and exocytosis (*Medina et al., 2011*; *Settembre et al., 2011*; *Giatromanolaki et al., 2015*; *Xu et al., 2020*), which may trigger cellular clearance. In fact, it is reported that TFEB overexpression promotes cellular clearance and ameliorates the phenotypes in a variety of neurodegenerative diseases, including Alzheimer's, Parkinson's, and Huntington's (*Sardiello et al., 2009*; *Tsunemi and La Spada, 2012*; *Decressac and Björklund, 2013*; *Polito et al., 2014*; *Ballabio, 2016*; *Napolitano and Ballabio, 2016*). Moreover, upregulation of TFEB has been reported to benefit lysosomal storage diseases (LSDs), such as Pompe disease (*Argüello et al., 2021*). However, it is not known whether upregulation of TFEB by genetic or pharmacological methods is sufficient to increase lysosomal function and alleviate NPC phenotypes in vitro or in vivo. If so, TFEB may be a putative target for NPC treatment and manipulating lysosomal function via small-molecule TFEB agonists may have broad therapeutic potential for NPC.

## Results

### Transgenic overexpression or pharmacological activation of TFEB reduces cholesterol accumulation in various human NPC1 cell models

TFEB and TFE3 are identified as key transcription factors regulating lysosome and autophagy biogenesis (*Sardiello et al., 2009*). To investigate the role of TFEB/TFE3 in cellular cholesterol levels in NPC1 cells, HeLa cells were treated with U18666A (2.5 μM, an inhibitor of the endosomal/lysosomal cholesterol transporter NPC1), a drug that has been widely used to induce NPC phenotype in cell models (*Poh et al., 2012*) (thereafter HeLa NPC1 cells represent U18666A-treated HeLa cells). Cellular cholesterol levels were measured by the well-known cholesterol-marker filipin (*Lu et al., 2015*). As shown in *Figure 1A and B*, in HeLa NPC1 cells overexpressing TFEB-mCherry or TFEB$^{S211A}$-mCherry (S211 non-phosphorylatable mutant, a constitutively active TFEB) (*Wang et al., 2015*), the cellular cholesterol levels (filipin) were significantly diminished compared to non-expressed or mCherry-only transfected NPC1 cells. In contrast, TFE3-GFP overexpression displayed no obvious reduction of cholesterol levels in HeLa NPC1 cells (*Figure 1—figure supplement 1*). Hence, TFEB but not TFE3 contributes to cholesterol reduction in NPC1 cells.

Pharmacological activation of TFEB is an emerging strategy for LSD treatment (*Levine and Kroemer, 2008*). We have previously identified a natural TFEB agonist sulforaphane (SFN), which is also an activator of cellular antioxidant NFE2L2/Nrf2 pathway, dramatically mitigating oxidative stress commonly associated with metabolic and age-related diseases, including NPC diseases (*Corssac et al., 2018*; *Li et al., 2021*, *Liu et al., 2021*). Based on our previous data that SFN (10–15 μM, 3–24 h) potently activates TFEB in various cell lines, promoting lysosomal function (*Li et al., 2021*), thus, in this study, we set SFN treatment dose to 15 μM and treatment time to 4–24 h according to individual experiment. We hypothesize that SFN may contribute to cholesterol clearance in NPC disease. To

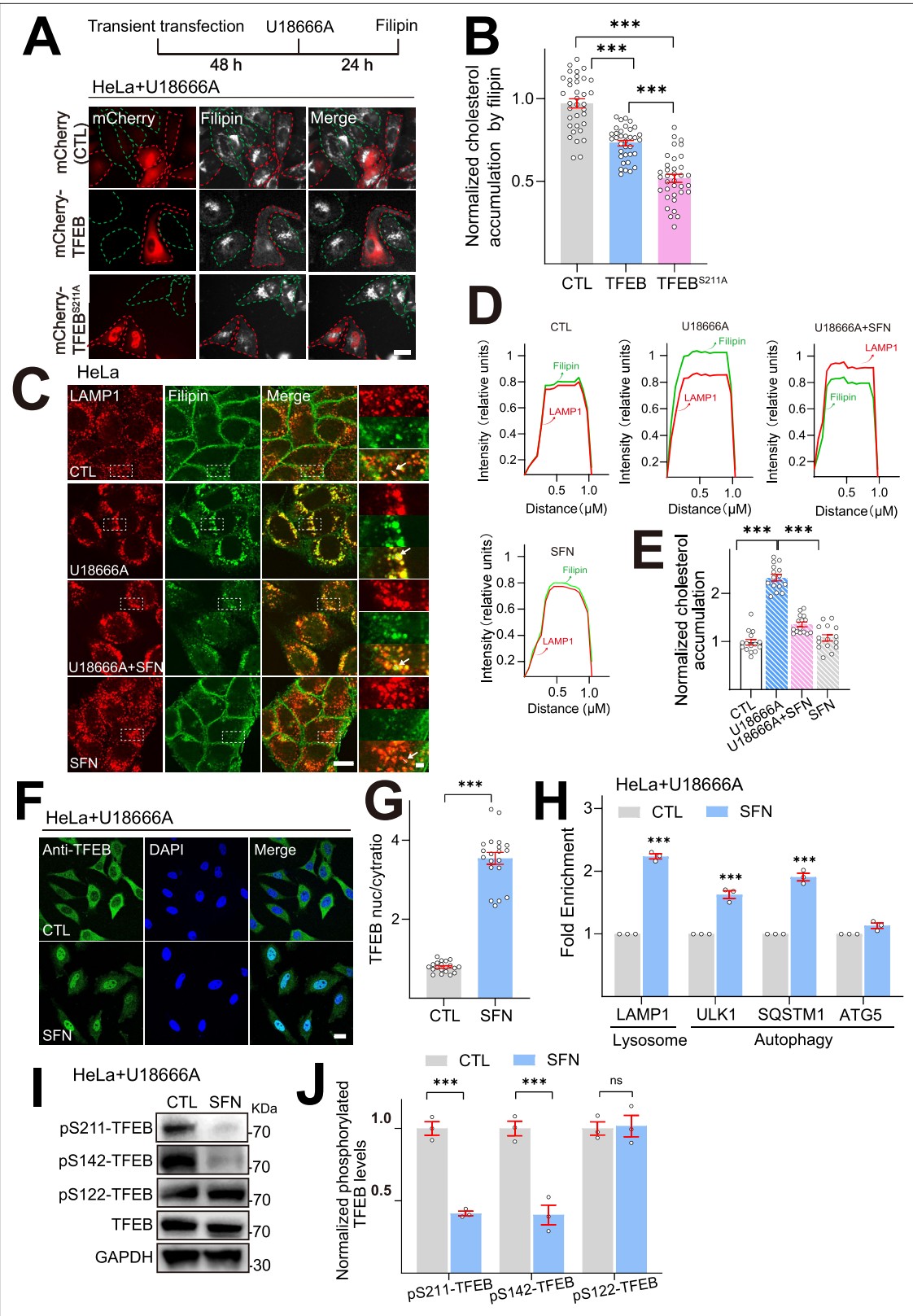

**Figure 1.** TFEB overexpression or pharmacological activation of TFEB ameliorates cholesterol accumulation in U18666A-induced HeLa NPC1 model. (**A**) Overexpression of TFEB/TFEB[S211A] reduced cellular cholesterol levels in U18666A-induced HeLa NPC1 model. Filipin staining of HeLa transiently transfected by mCherry, mCherry-TFEB and mCherry- TFEB[S211A] plasmid for 48 h, followed by U18666A (2.5 μM) for 24 h. Overlay phase-contrast images are shown together with the red (mCherry-TFEB/TFEB[S211A]) and white (filipin). In each image, the red circles point to the successfully transfected cells,

*Figure 1 continued on next page*

*Figure 1 continued*

green circles represent the untransfected cells. Scale bar, 20 µm. (**B**) Quantification of cholesterol accumulation from (**A**). N = 30 randomly selected cells from n = 3 independent experiments. (**C**) Sulforaphane (SFN) reduces lysosomal (LAMP1) cholesterol accumulation (filipin) in HeLa NPC1 cells. HeLa cells were exposed to U18666A (2.5 µM) in the absence or the presence of SFN (15 µM) for 24 h. Each panel shows fluorescence images taken by confocal microscopes. The red signal is LAMP1-mCherry driven by stable transfection, and the green signal is filipin. (**D**) Each panel shows the fluorescence intensity of a line scan (white line on the blown-up image) through the double-labeled object indicated by the white arrow. Scale bar, 20 µm or 2 µm (for zoom-in images). (**E**) Quantification of cholesterol levels shown in (**C**). N = 15 randomly selected cells from n = 3 experiments. (**F**) SFN (15 µM, 24 h) induced TFEB nuclear translocation in HeLa NPC1 cells. Nuclei were counterstained with DAPI (blue). Scale bar, 20 µm. (**G**) Average ratios of nuclear vs. cytosolic TFEB immunoreactivity shown in (**F**). N = 20 randomly selected cells from n = 3 experiments. (**H**) SFN-induced mRNA expression of TFEB target genes in HeLa NPC1 model. HeLa cells were cotreated with U18666A (2.5 µM) and SFN (15 µM) for 24 h (n = 3). (**I**) Western blot analysis of TFEB phosphorylation by SFN (15 µM, 24 h) in HeLa NPC1 cells. (**J**) Quantification of ratios of pS211-, pS142-, pS122-TFEB vs. total TFEB as shown in (**I**) (n = 3). For all the panels, average data are presented as mean ± s.e.m.; ***p<0.001.

The online version of this article includes the following source data and figure supplement(s) for figure 1:

**Source data 1.** Original western blots for *Figure 1I*, indicating the relevant bands and treatments.

**Source data 2.** Original files for western blot analysis displayed in *Figure 1I*.

**Figure supplement 1.** The effect of TFE3 overexpression on lysosomal cholesterol levels in HeLa NPC1 model.

**Figure supplement 2.** Filipin signal is not colocalized with ER marker calnexin in HeLa NPC1 cells.

**Figure supplement 3.** Sulforaphane (SFN)-induced TFEB nuclear translocation via a ROS-Ca$^{2+}$-calcineurin pathway in HeLa NPC1 cells.

**Figure supplement 4.** Western blot analysis of the phosphorylation status of MTOR and RPS6KB1 by sulforaphane (SFN).

**Figure supplement 4—source data 1.** Original western blots for *Figure 1—figure supplement 4A*, indicating the relevant bands and treatments.

**Figure supplement 4—source data 2.** Original files for western blot analysis displayed in *Figure 1—figure supplement 4A*.

**Figure supplement 5.** Western blot analysis of NPC1 expression in human NPC1 patient fibroblasts.

**Figure supplement 5—source data 1.** Original western blots for *Figure 1—figure supplement 5*, indicating the relevant bands and treatments.

**Figure supplement 5—source data 2.** Original files for western blot analysis displayed in *Figure 1—figure supplement 5*.

**Figure supplement 6.** Sulforaphane (SFN) induced Nrf2 nuclear translocation in NPC cells.

evaluate this hypothesis, we first examined the effect of SFN on cellular cholesterol levels in HeLa NPC1 cell model using filipin staining. As shown in *Figure 1C and D*, U18666A treatment significantly increased filipin signals in bright perinuclear granules, which were well co-localized with lysosomal marker LAMP1, but not with ER marker calnexin in HeLa cells (*Figure 1—figure supplement 2*), suggesting that U18666A-induced cholesterol accumulation presents in the late endosome/lysosome compartment, but not in ER. Moreover, when HeLa NPC1 cells were further challenged with SFN (15 µM, 12–24 h), a dramatic reduction of cholesterol accumulation (filipin intensity) by more than 30% was observed in lysosomes (*Figure 1C–E*, *Figure 1—figure supplement 2*). These results confirmed that U18666A interferes with the egress of free cholesterol from endosomes/lysosomes as previously reported by others (*Lange and Steck, 1994*; *Davis et al., 2021*) and SFN reduces lysosomal cholesterol in NPC1 cells.

In our previous study, we identified that SFN activates TFEB via a ROS-Ca$^{2+}$- calcineurin-mediated but MTOR (mechanistic target of rapamycin kinase)-independent mechanism (*Li et al., 2021*). Thus, we further validated this mechanism of SFN in HeLa NPC1 cell model by performing immunofluorescence experiments. In HeLa NPC1 cells, SFN (15 µM, 24 h) induced robust TFEB translocation from the cytosol to the nucleus (*Figure 1F and G*). SFN treatment (15 µM, 24 h) also resulted in an increase in the mRNA levels of TFEB downstream genes, including genes required for autophagy (*ULK1*, *SQSTM1*, and *ATG5*) and lysosome biogenesis-*LAMP1*, using quantitative real-time PCR (Q-PCR) (*Figure 1H*). Consistent with previous findings, BAPTA-AM (a membrane-permeable Ca$^{2+}$ chelator), FK506 and cyclosporin A (CsA) (calcineurin inhibitors), and N-acetylcysteine (NAC, a ROS scavenger) can robustly block SFN-induced TFEB nuclear translocation in HeLa NPC1 cells (*Figure 1—figure supplement 3*), suggesting that in NPC1 cells SFN induces TFEB nuclear translocation via a ROS-Ca$^{2+}$-calcineurin-dependent mechanism A key mechanism of TFEB activation is Ca$^{2+}$-dependent dephosphorylation of TFEB by protein phosphatases (*Napolitano et al., 2018*). We next investigated the specific phosphorylated site on TFEB by SFN in NPC1 cells. Following 24 h exposure to SFN, S211, and S142-TFEB phosphorylation were significantly decreased (*Figure 1I and J*), while S122 phosphorylation was not affected. These results indicate that TFEB is dephosphorylated at S211 and S142

residues by SFN in HeLa NPC1 model. TFEB nuclear shuffling is regulated by the activity of MTOR, which is regulated by its phosphorylation status (*Martina et al., 2012*). SFN reportedly activates TFEB in a mTOR-independent manner (*Li et al., 2021*). Consistently, no significant inhibition of phosphorylated MTOR (p-MTOR) or RPS6KB1 (p-RPS6KB1) can be observed in HeLa NPC1 cells treated with SFN (15 µM, 24 h) (*Figure 1—figure supplement 4*). Therefore, in HeLa NPC1 cells SFN-induced TFEB activation is unlikely to be mediated by MTOR inhibition. We also examined the possibility of the direct effect of SFN on NPC1 by western blotting. As shown in *Figure 1—figure supplement 5*, SFN (15 µM, 24 h) treatment did not affect NPC1 expression in human NPC1-patient fibroblasts. Given the well-known role of SFN as an NFE2/Nrf2 inducer, we also validated whether SFN can induce Nrf2 activation in NPC cells. As shown in *Figure 1—figure supplement 6*, SFN (15 µM, 24 h) induced robust Nrf2 nuclear translocation from the cytosol to the nucleus in HeLa NPC1 and NPC primary mouse embryonic fibroblasts (MEF) cells.

Next, we verified the effect of SFN in another NPC1 cell model by knockdown (KD) *NPC1* with specific siRNA (*Liao et al., 2015*; *Höglinger et al., 2019*). In *NPC1* KD HeLa cells with more than 80% knockdown efficiency (*Figure 2A*), SFN can induce TFEB nuclear translocation (*Figure 2B and C*) and upregulate its downstream gene expression (*Figure 2D*). Notably, SFN (15 µM, 24 h) treatment significantly reduced cholesterol levels in *NPC1* KD cells (*Figure 2E and F*). Similar results were observed in human NPC1-patient fibroblast cells. SFN treatment robustly promoted TFEB nuclear translocation (*Figure 2G and H*) and cholesterol clearance (*Figure 2I and J*). Taken together, these experiments demonstrate that pharmacological activation of TFEB activation by SFN could promote cellular cholesterol clearance in various NPC in vitro models.

## TFEB is required in SFN -promoted cholesterol clearance in NPC1 cells

We next investigated whether TFEB is required for SFN-promoted cholesterol clearance using two strategies. On the one hand, HeLa cells were treated with the siRNAs specifically against TFEB or transiently overexpressed with mCherry-TFEB. The efficiency of siRNA KD and overexpression was evaluated by western blot (*Figure 3A*). In the si*TFEB*-transfected HeLa NPC1 cells, SFN-promoted (15 µM, 24 h) cholesterol clearance was almost abolished (*Figure 3B and C*). In contrast, in scrambled siRNA-transfected NPC1 cells, the cholesterol levels were significantly reduced by more than 30% by SFN. Notably, TFEB overexpression-induced reduction of cholesterol can be further boosted about 20% by SFN treatment (*Figure 3B and C*). On the other hand, HeLa *TFEB* KO cells were constructed by the CRISPR/Cas9 tool (*Figure 3D*). Consistently, SFN failed to diminish cholesterol in HeLa *TFEB* KO cells, whereas re-expression of a recombinant TFEB restored the cholesterol clearance by SFN (*Figure 3E and F*). Collectively, these data indicate that TFEB is specifically required for lysosomal cholesterol reduction upon SFN treatment.

## SFN promotes lysosomal exocytosis and biogenesis in human NPC1 models in a TFEB-dependent manner

We then studied the mechanism by which SFN led to diminished cellular cholesterol in NPC1 cells. Considering the established role of TFEB activation in lysosomal exocytosis, we verified the hypothesis that SFN-induced cholesterol clearance through upregulation of lysosomal exocytosis using surface LAMP1 immunostaining (*Figure 4A*). When lysosomal exocytosis processes, luminal lysosomal membrane proteins can be detected on the extracellular side of the plasma membrane (PM) by measuring surface expression of LAMP1 (lysosomal-associated membrane protein 1), a resident marker protein of late endosomes and lysosomes (referred to as 'lysosomes' for simplicity hereafter) with a monoclonal antibody against a luminal epitope of LAMP1 (*Reddy et al., 2001*). After incubation with SFN (15 µM) for 24 h, HeLa NPC1 cells exhibited a dramatic increase in the signal of LAMP1 staining in the PM (surface LAMP1 signal colocalized with DiO, a PM marker) compared to untreated control cells (*Figure 4A and B*), suggesting an increase of lysosomal exocytic process. Likewise, SFN promoted lysosomal exocytosis in primary macrophage cells (*Figure 4—figure supplement 1*). In contrast, this increase of surface LAMP1 signal by SFN was significantly reduced in *TFEB* KO cells (*Figure 4—figure supplement 1*), demonstrating that SFN-induced lysosomal exocytosis is TFEB-dependent.

A direct consequence of lysosomal exocytosis is the release of lysosomal contents into the cell culture medium (*Rodríguez et al., 1997*). We then quantified the release of free cholesterol/

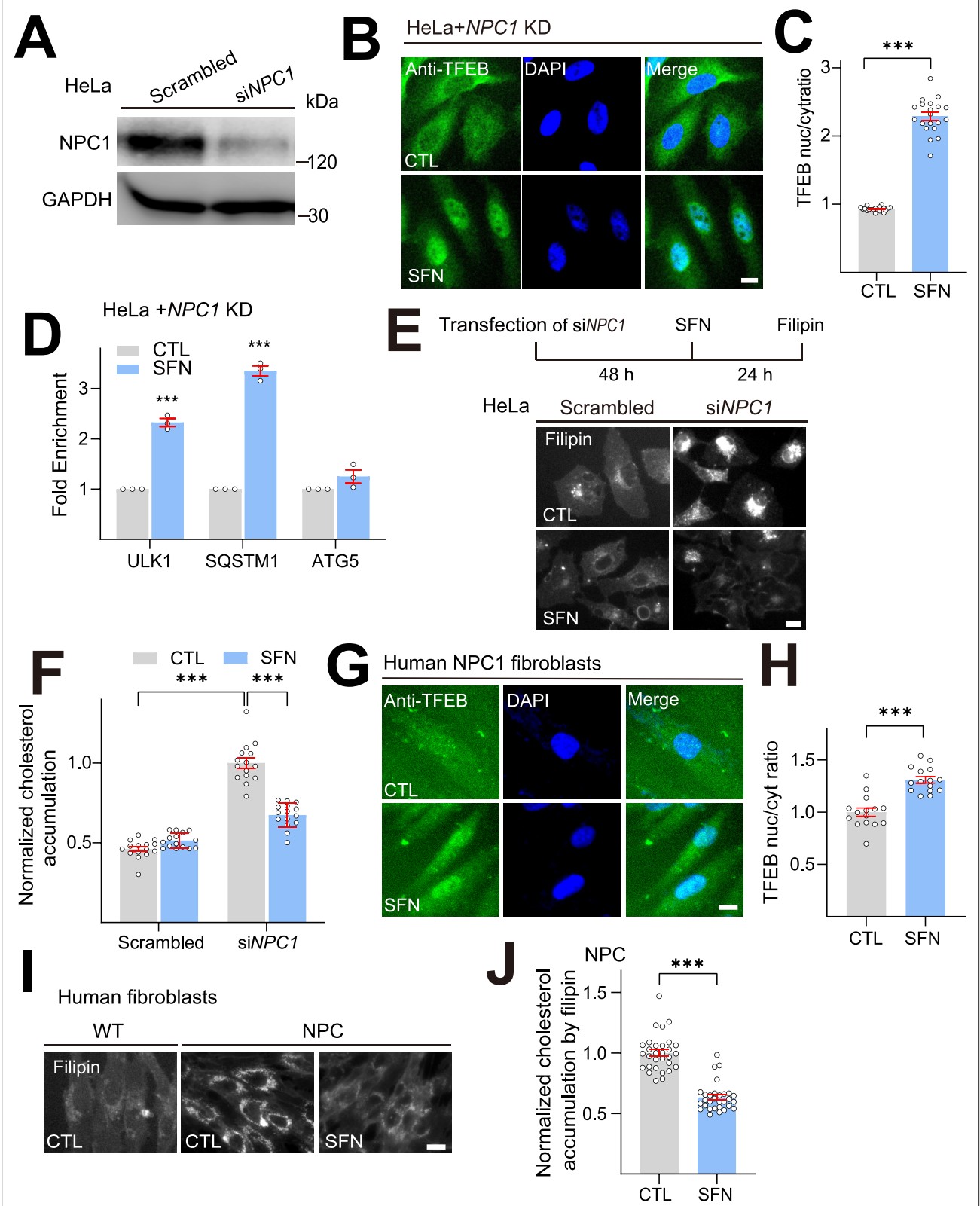

**Figure 2.** Sulforaphane (SFN) promotes cholesterol clearance in various human NPC1 cell models. (**A**) Western blot analysis of the KD efficiency of a specific NPC1-targeting siRNA in HeLa cells (n = 3 independent repeats). (**B**) SFN-induced TFEB nuclear translocation in NPC1 KD HeLa cells. Detection of TFEB immunoreactivity in HeLa cells transiently transfected with si*NPC1* for 48 h, followed by SFN (15 μM) treatment for 24 h. Scale bar, 20 μm. (**C**) Average ratios of nuclear vs. cytosolic TFEB immunoreactivity shown in (**B**). N = 20 randomly selected cells from three independent repeats. (**D**) In NPC1

*Figure 2 continued on next page*

*Figure 2 continued*

KD HeLa cells, SFN (15 µM, 24 h) upregulated expression of TFEB target genes (n = 3 independent repeats). (**E**) SFN promoted cholesterol clearance in NPC1 KD HeLa cells. HeLa cells were transiently transfected by si*NPC1* for 48 h, followed by SFN (15 µM) treatment for 24 h. Scale bar, 20 µm. (**F**) Quantification of cholesterol accumulation in NPC1 KD HeLa cells shown in (**E**). N = 15 randomly selected cells from three independent repeats. (**G**) SFN (15 µM, 24 h)-induced TFEB nuclear translocation in human NPC1 fibroblasts. Nuclei were counterstained with DAPI (blue). Scale bar, 20 µm. (**H**) Quantification of nuclear vs. cytosolic TFEB ratio as shown in (**G**). N = 20 randomly selected cells from at least three independent experiments. (**I**) SFN promoted cholesterol clearance in human NPC1-patient fibroblasts. Human NPC1 fibroblasts were treated with SFN (15 µM, 24 h) and filipin staining was carried out. Scale bar, 20 µm. (**J**) Quantification of cholesterol accumulation as shown in (**I**). N = 30 randomly selected cells from three independent repeats. For all the panels, average data are presented as mean ± s.e.m.; \*\*\*p<0.001.

The online version of this article includes the following source data for figure 2:

**Source data 1.** Original western blots for *Figure 2A*, indicating the relevant bands and treatments.

**Source data 2.** Original files for western blot analysis displayed in *Figure 2A*.

cholesteryl ester into the medium upon SFN treatment in HeLa NPC1 cells using the Cholesterol/Cholesteryl Ester assay Kit. This assay allows to detect total cholesterol or free cholesterol in the presence or absence of cholesterol esterase; the levels of cholesteryl ester are determined by subtracting the levels of free cholesterol from total cholesterol. As shown in *Figure 4C*, the levels of free cholesterol released into the medium treated were significantly increased with SFN (15 µM, 24 h) treatment compared to untreated control, suggesting that SFN can stimulate the release of free cholesterol into the medium. Notably, the difference of released cholesteryl ester with or without SFN treatment has not been observed, and the percentage of cholesteryl ester in total cholesterol released to medium was less than 10% under all conditions tested. Meanwhile, we further examined the effect of SFN on the release of lysosomal enzymes β-hexosaminidase (NAGase) and acid phosphatase (ACP). In HeLa NPC1 cells treated with SFN (15 µM, 24 h), significantly higher levels of lysosomal enzymes (NAGases and ACP) were detected in the medium compared with untreated controls, but not in *TFEB* KO cells (*Figure 4D*). Taken together, these results indicate that SFN induces an active movement of lysosomes toward the PM and increases lysosomal exocytosis in a TFEB-dependent manner.

Previously we reported that SFN could increase lysosome biogenesis and regulate lysosomal function, which contribute to ROS reduction in NPC models (*Li et al., 2021*). We then studied whether SFN activates lysosomal machinery in NPC1 cells. In HeLa NPC1 cells, SFN (15 µM, 12 h) treatment significantly increased the immunofluorescence intensity of LAMP1 (*Figure 4E and F*). Likewise, similar results were observed in NPC1 KD cells (*Figure 4—figure supplement 2*). Lysosomal enzymes operate better under acidic conditions, and the degradation-active lysosomes can be tracked using LysoTracker, a fluorescent acidotropic probe (*Li et al., 2019*). We observed significant increases of LysoTracker staining in HeLa NPC1 cells following 12 h treatment with SFN (15 µM) (*Figure 4G and H*) as well as in NPC1 KD cells (*Figure 4—figure supplement 2*), yet LAMP1 staining was also increased (*Figure 4E and F*, *Figure 4—figure supplement 2*). Thus, lysosomal pH was more accurately determined using a ratiometric dye pHrodo Green Dextran. When the fluorescence ratios (pHrodo Green Dextran/CF555) were calibrated to pH values, we found that SFN treatment induced significant lysosomal hyperacidity (*Figure 4I and J*). Collectively, these results suggest that SFN promotes lysosome function and biogenesis in human NPC1 model cells, consistent with our previous report.

Notably, we then examined the TFEB expression in two NPC cell lines (HeLa NPC1 cells and NPC1-patient fibroblast cells). As shown in *Figure 4—figure supplement 3*, TFEB expression levels in both NPC cell models were significantly decreased compared to WT, indicating that TFEB expression may be inhibited in NPC cells. Interestingly, the basal levels of lysosome biogenesis in NPC cells were comparable with WT cells (*Figure 4D–G*, *Figure 4—figure supplement 2*), suggestive of compensatory changes caused by TFEB downregulation. Taken together, excessive activation of TFEB in NPC cells can be targeted for cholesterol clearance via upregulation of lysosomal function and biogenesis.

## SFN alleviates cholesterol accumulation in primary *Npc1*<sup>-/-</sup> MEF cells

To address the possible relevance of the SFN/TFEB axis in NPC pathology of mice experiments, we then investigated whether SFN is sufficient to reduce cholesterol and regulates lysosomal function via TFEB activation in primary murine cells. Primary MEFs were freshly prepared from *Npc1*<sup>-/-</sup> mice (from Jackson's laboratory). SFN (15 µM) treatment for 24 h dramatically increased TFEB nuclei signal in the *Npc1*<sup>-/-</sup> MEF cells (*Figure 5A and B*). Next, we analyzed the effect of SFN on cholesterol clearance in

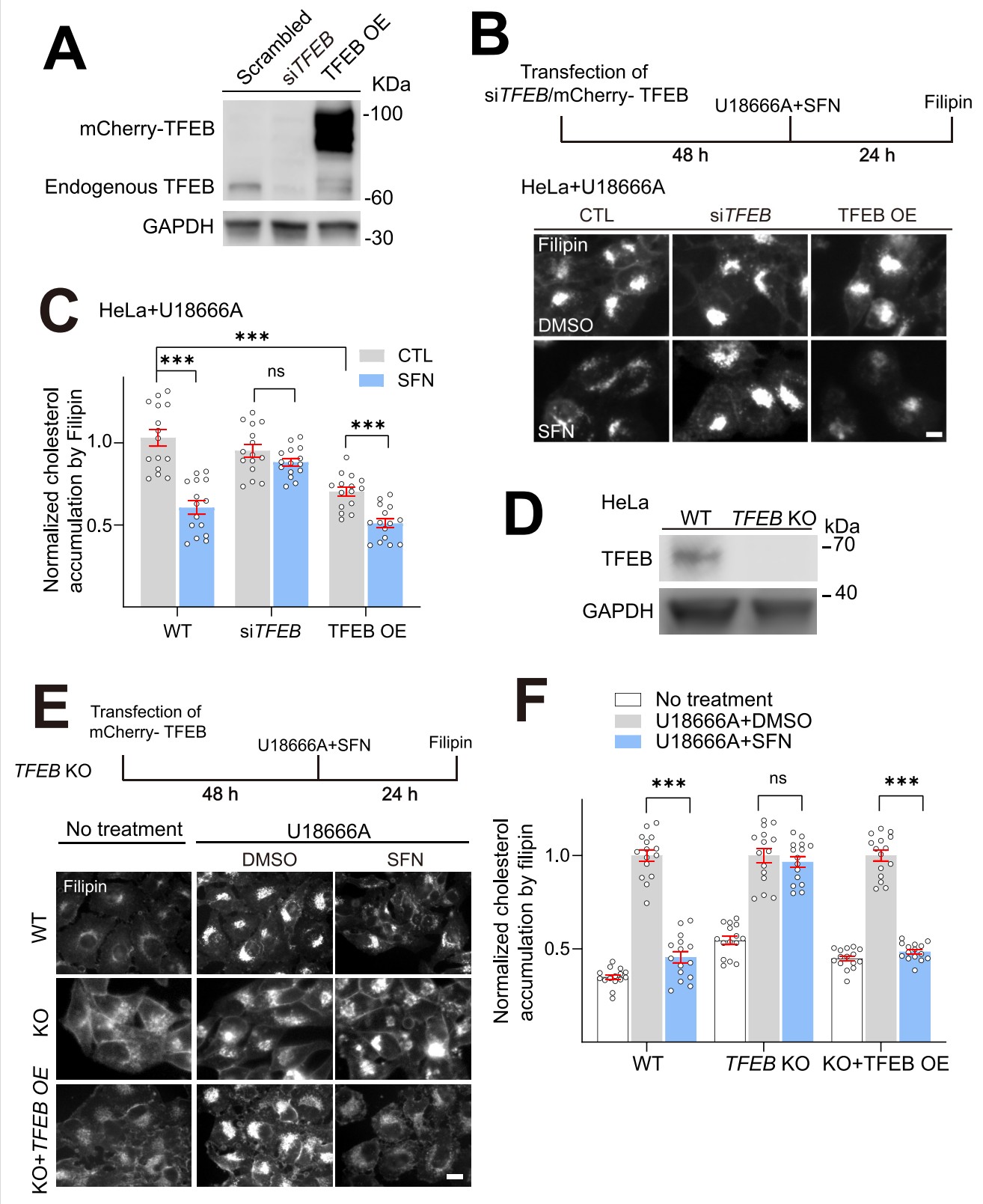

**Figure 3.** TFEB is required for sulforaphane (SFN)-promoted cholesterol clearance. (**A**) Western blot analysis of the efficiency of si*TFEB* KD and mCherry-TFEB OE in HeLa cells. (**B**) HeLa cells were transfected with si*TFEB* or mCherry-TFEB for 48 h, followed by cotreatment with U18666A (2.5 μM) and SFN (15 μM) for 24 h and cholesterol accumulation was analyzed by Filipin assay. Scale bar, 20 μm. (**C**) Quantification of cholesterol levels as shown in (**B**). N = 15 randomly selected cells from three independent repeats. (**D**) Western blot analysis of the efficiency of *TFEB* KO in HeLa cells. (**E**) HeLa,

*Figure 3 continued on next page*

*Figure 3 continued*

HeLa *TFEB* KO, and HeLa *TFEB* KO cells transient expressing mCherry-TFEB (TFEB OE, 48 h) were cotreatment with U18666A (2.5 μM) and SFN (15 μM) for 24 h, and cholesterol levels were analyzed by filipin assay. Scale bar, 20 μm. (**F**) Quantification analysis of cholesterol accumulation as shown in (**E**). N = 15 randomly selected cells from at least three independent experiments. Average data are presented as mean ± s.e.m.; ***p<0.001.

The online version of this article includes the following source data for figure 3:

**Source data 1.** Original western blots for *Figure 3A*, indicating the relevant bands and treatments.

**Source data 2.** Original files for western blot analysis displayed in *Figure 3A*.

**Source data 3.** Original western blots for *Figure 3D*, indicating the relevant bands and treatments.

**Source data 4.** Original files for western blot analysis displayed in *Figure 3D*.

*Npc1*[-/-] MEF cells. SFN (15 μM) treatment for 72 h exhibited substantial cholesterol reduction, whereas SFN treatment for 24 h showed a relatively weaker cholesterol clearance in MEF cells (*Figure 5C and D*) compared with human NPC1 cells (*Figure 1C*), suggesting that human NPC cells are more sensitive to SFN treatment compared to mouse NPC cells. We further tested whether SFN promotes lysosomal biogenesis and function in *Npc1*[-/-] MEF cells. Following 24 h treatment with SFN (15 μM), a significant increase of LAMP1 staining (*Figure 5E and F*) and LysoTracker intensity (*Figure 5G and H*) were observed in *Npc1*[-/-] MEF cells. Collectively, these results suggest that SFN regulates TFEB-mediated lysosomal function axis and promotes cellular cholesterol clearance in NPC MEF cells.

## SFN alleviates the loss of Purkinje cells and body weight in *Npc1*[-/-] mice

Considering that SFN promotes lysosomal cholesterol clearance in both human and murine NPC1 cell models (*Figures 1, 2 and 5*) and reportedly penetrates blood–brain barrier (BBB) (*Kim et al., 2013*; *Mao et al., 2019*; *Tavakkoli et al., 2019*). We next investigated whether SFN targets/activates TFEB in brain. 4-week-old BALB/cJ mice were intraperitoneally injected with SFN (50 mg/kg) or vehicle for 12 h, brain tissues, including cerebellum and hippocampus, were collected, and pS211-TFEB/TFEB levels were measured by western blotting. As shown in *Figure 6A and B*, we observed a significant decrease of pS211-TFEB protein in brain tissues with SFN treatment compared to vehicle, suggesting that TFEB in the brain was directly targeted by SFN treatment. This is the first time that SFN was shown to directly active TFEB in the brain. We then evaluated the in vivo therapeutic efficacy of SFN. *Npc1*[-/-] mice (4-week-old) were treated with SFN (30 or 50 mg/kg) by daily intraperitoneal injection for 4 weeks. Purkinje cells located in the cerebellum are the most susceptible to NPC1 loss and exhibit a significant selective loss in the anterior part of the cerebellum (*Sarna et al., 2003*). Purkinje cells in cerebellum sections were stained by calbindin and quantified by recording the number of surviving cells in lobules/mm of Purkinje cell layer. As shown in *Figure 6C and D*, little survival of Purkinje cells in vehicle-treated *Npc1*[-/-] cerebellum, in contrast, daily injection of SFN (50 mg/kg) in *Npc1* mice prevented a degree of Purkinje cell loss, particularly in the lobule IV/V of cerebellum. Body weight is another important indicator of therapeutic efficacy in *Npc1* mice. Typically, *Npc1* mice weight plateaus at 6–7 weeks of age, and then progressively declines. Notably, we observed that daily intraperitoneal injection of SFN (50 mg/kg) exhibited a significant improvement in weight loss of *Npc1* mice (*Figure 6E*). However, SFN treatment has no effect on the liver and spleen enlargement of *Npc1* mice (data not shown). Collectively, our results demonstrated that pharmacological activation of TFEB by small-molecule agonist can mitigate NPC neuropathological symptoms in vivo.

## Discussion

We have demonstrated in the current study that genetic overexpression of TFEB, but not TFE3, can dramatically mitigate cholesterol accumulation in NPC cells. Pharmacological activation of TFEB by SFN, a previously identified TFEB agonist (*Li et al., 2021*), significantly promoted cholesterol clearance in human and mouse NPC cells, while genetic inhibition (KO) of TFEB blocked SFN-induced cholesterol clearance. This clearance effect exerted by SFN was associated with upregulated lysosomal exocytosis and biogenesis (*Figure 7*). Notably, SFN is reportedly BBB-permeable, assuring a good candidate for efficient delivery to the brain, which is essential for targeting neurodegenerative phenotypes in neurological diseases including NPC. In the NPC mouse models, SFN exhibits in vivo efficacy of suppressing the loss of Purkinje cells and maintaining body weight. Hence, genetically or

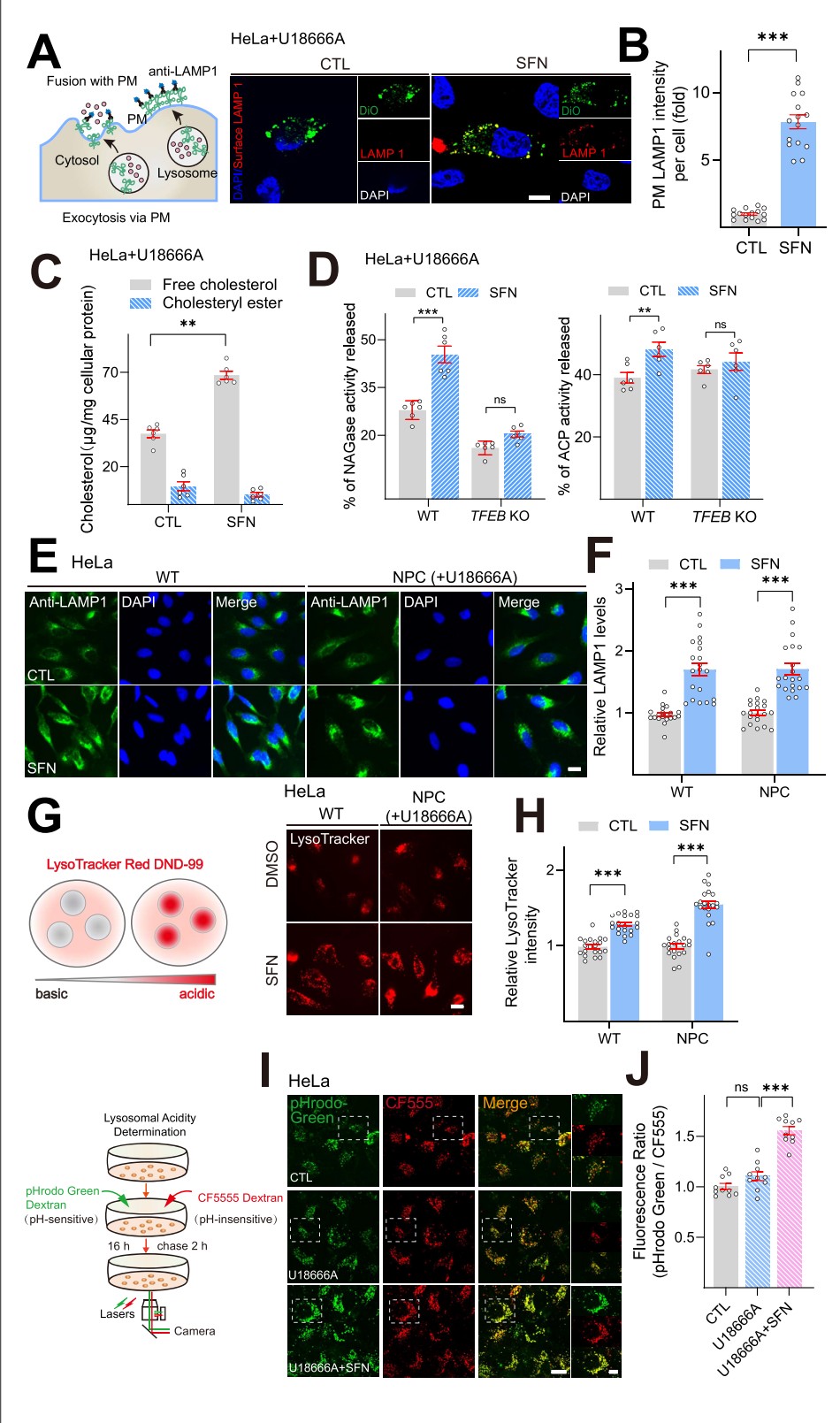

**Figure 4.** Sulforaphane (SFN) promotes lysosomal exocytosis and biogenesis in NPC1 cell models. (**A**) Confocal microscopy images showing the exposure of LAMP1 on the plasma membrane (PM) in nonpermeabilized HeLa NPC1 cells treated with SFN (15 μM) for 24 h using an antibody against LAMP1 luminal portion. Nuclei were counterstained with DAPI (blue). Scale bar, 20 μm. (**B**) Quantitative analysis of LAMP1 levels on the PM in HeLa

*Figure 4 continued on next page*

*Figure 4 continued*

NPC1 cells shown in (**A**). Bars represent the fold increase of LAMP1 fluorescence on PM in SFN-treated cells. N = 15 randomly selected cells from three independent repeats. (**C**) SFN increased the release of free cholesterol into the medium. HeLa cells were cotreated with U18666A (2.5 μM) and SFN (15 μM) for 24 h, and then examined for cholesterol. The levels of cholesterol in the medium or cell lysates were measured by cholesterol assay in a reaction mixture with (measuring total cholesterol content) or without (measuring free cholesterol content) cholesterol esterase enzyme (n = 6 independent repeats). (**D**) SFN increased the release of lysosomal enzyme NAGases and ACP in HeLa NPC1 cells. HeLa cells were cotreated with U18666A (2.5 μM) and SFN for 24 h, and the activities of NAGases and ACP were analyzed in the medium and cell lysates (n = 6 independent repeats). (**E**) LAMP1 staining in HeLa cells upon U18666A treatment (2.5 μM) in the presence and absence of SFN (15 μM). Nuclei were counterstained with DAPI (blue). Scale bar, 20 μm. (**F**) Quantification analysis of LAMP1 immunofluorescence shown in (**E**). N = 20 randomly selected cells from at least three independent experiments. (**G**) Effects of SFN on lysosome acidity. HeLa cells were treated with 2.5 μM U18666A (24 h) in the presence and absence of 15 μM SFN (12 h) and lysosomal pH was analyzed by LysoTracker Red DND-99 (50 nM). Scale bar, 20 μm. (**H**) Quantification of LysoTracker intensity shown in (**G**). N = 20 randomly selected cells from at least three independent experiments. (**I**) Effects of SFN on lysosomal acidity using a ratiometric pH dye. HeLa cells were treated with U18666A (2.5 μM) in the presence and absence of SFN (15 μM), lysosomal pH was determined using a ratiometric pH dye combination (pHrodo Green dextran and CF555 dextran). Scale bar, 20 μm or 2 μm (for zoom-in images). (**J**) Quantification analysis of lysosomal pH shown in (**I**). Randomly selected cells from at least three independent experiments. For all the panels, data are presented as mean ± s.e.m.; **p<0.01, ***p<0.001.

The online version of this article includes the following source data and figure supplement(s) for figure 4:

**Figure supplement 1.** Sulforaphane (SFN) induces lysosomal exocytosis in a TFEB-dependent manner.

**Figure supplement 2.** SFN promotes lysosomal biogenesis in *NPC1* KD HeLa cells.

**Figure supplement 3.** TFEB expression is downregulated in NPC1 cells.

**Figure supplement 3—source data 1.** Original western blots for *Figure 4—figure supplement 3*, indicating the relevant bands and treatments.

**Figure supplement 3—source data 2.** Original files for western blot analysis displayed in *Figure 4—figure supplement 3*.

**Figure supplement 4.** The cytotoxic effects of sulforaphane (SFN) on various cell lines.

pharmacologically targeting TFEB may represent a promising approach to treat NPC and manipulating lysosome function with small-molecule TFEB agonists may have broad therapeutic potentials.

The MiT/TFE family contains four factors: MITF, TFEB, TFE3, and TFEC, which share an identical basic region for DNA binding, and highly similar HLH and Zip regions for dimerization (**Haq and Fisher, 2011**). Many of the mechanistic insights into MiT regulation have been focused on TFEB and TFE3, which shares some overlapping functions, Surprisingly, in this study we found that overexpression of TFEB, but not TFE3, alleviated lysosomal cholesterol accumulation in NPC cells (**Figure 1A and B Figure 1—figure supplement 1**). Moreover, studies reported that only TFEB overexpression, but not other MiT members, upregulates lysosomal gene expression (**Sardiello et al., 2009**). Thus, these results suggest that the functions exerted by TFEB and TFE3 in NPC may appear to be specialized.

As a proof of the role of TFEB activation in NPC, pharmacological activation of TFEB by SFN, a natural small-molecule TFEB agonist, promotes a dramatic lysosomal cholesterol-lowering effect in several genetic and pharmacological NPC cell models (**Figures 1C and 5C**). The concentration of SFN used in this study has no cytotoxicity toward the cell lines we used (**Figure 4—figure supplement 4**). SFN-induced lysosomal exocytosis and the increased population of lysosomes ready to fuse with the PM contribute to the cholesterol clearance by SFN (**Figure 4A–G**). Previously we have shown that SFN can mitigate oxidative stress via a ROS-$Ca^{2+}$-calcineurin-TFEB-mediated lysosomal function and autophagy flux (**Li et al., 2021**). SFN, an electrophilic compound enriched in cruciferous vegetables such as broccoli, is a known potent inducer of NFE2L2/NRF2 in various cell types including NPC cells (**Figure 1—figure supplement 6**), a transcriptional factor that controls the expression of multiple detoxifying enzymes through antioxidant response elements (AREs) (**Yamamoto et al., 2018**). Notably, the promoter region of the *Nfe2l2* gene harbors a CLEAR site, and TFEB activation can upregulate the expression of *Nfe2l2* (**Mansueto et al., 2017**). Hence, TFEB, together with NFE2L2, functions as a key regulator of cellular redox. Oxidative stress is a major feature of NPC and has been attributed to neuronal damage, leading to the pathogenesis and progression of NPC (**Vázquez**

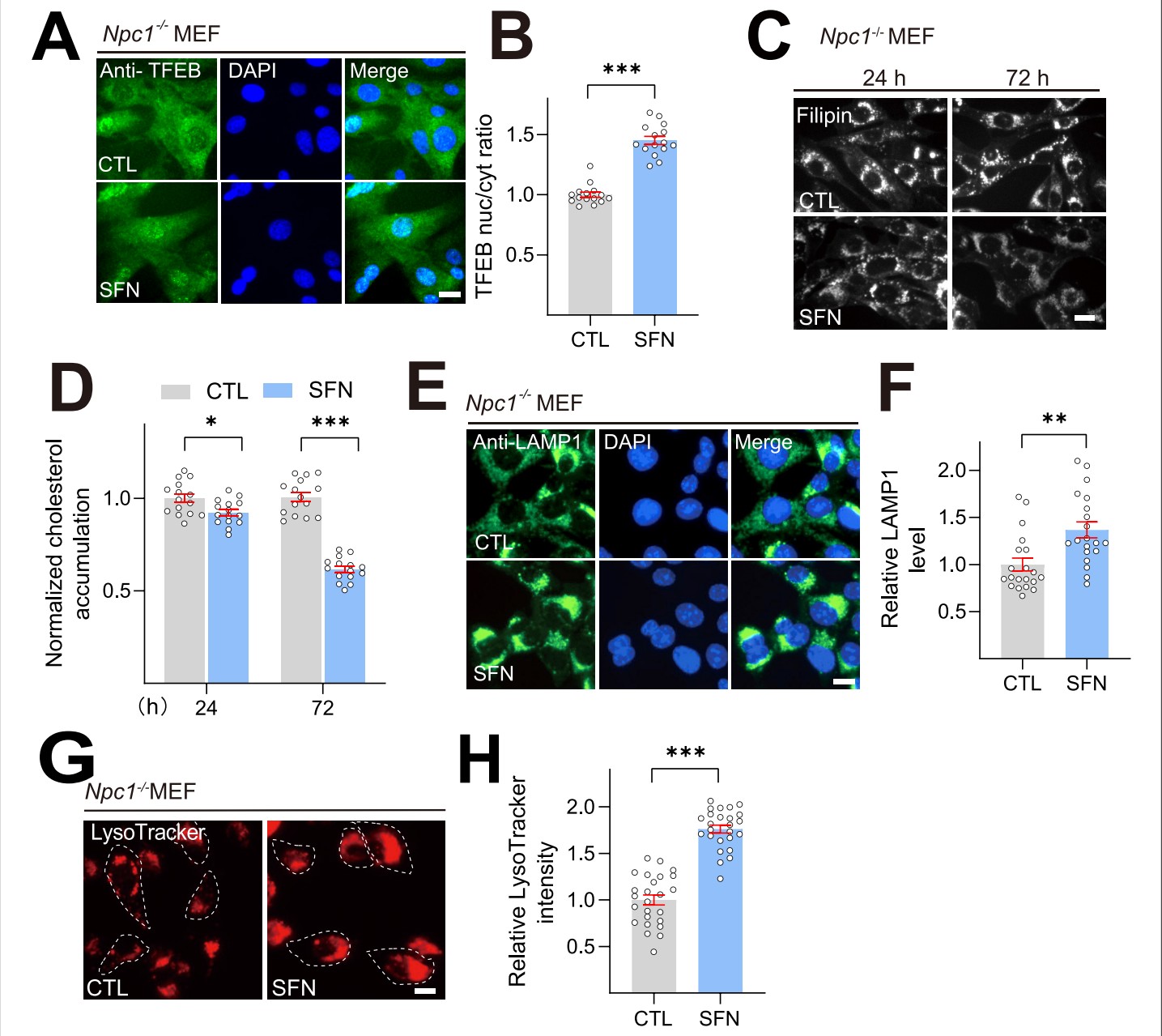

**Figure 5.** Sulforaphane (SFN) ameliorates cholesterol accumulation in *Npc1⁻/⁻* mouse embryonic fibroblast (MEF) cells. (**A**) SFN (15 µM) treatment induced TFEB nuclear translocation in *Npc1* MEF cells. Nuclei were counterstained with DAPI (blue). Scale bar, 20 µm. (**B**) Average ratios of nuclear vs. cytosolic TFEB immunoreactivity shown in (**A**). N = 20 from three independent repeats. (**C**) SFN (15 µM, 24–72 h) reduced cholesterol accumulation in *Npc1* MEF cells by filipin assay. Scale bar, 20 µm. (**D**) Quantification analysis of cholesterol accumulation levels shown in (**C**). N = 15 randomly selected cells from at least three independent experiments. (**E**) Effects of SFN (15 µM, 12 h) on the intensity of LAMP1 in *Npc1* MEF cells. Scale bar, 20 µm. (**F**) Quantification of LAMP1 intensity shown in (**E**). N = 20 randomly selected cells from at least three independent experiments. (**G**) Effects of SFN (15 µM, 12 h) on lysosome acidity in MEF cells. Scale bar, 20 µm. (**H**) Quantification analysis of LysoTracker intensity shown in (**G**). N = 20 randomly selected cells from at least three independent experiments. For all the panels, data are presented as mean ± s.e.m.; *p<0.05, **p<0.01, ***p<0.001.

*et al., 2012*). Elevated oxidative stress has been observed in the brain of NPC patient and mice (*Smith et al., 2009*; *Zampieri et al., 2009*). NPC patients also show decreased antioxidant capacity (expressed as Trolox equivalents) and diminished activity of different antioxidant enzymes, which indicates a decrease in antioxidant defenses (*Fu et al., 2010*). Hence, the protective effect of SFN against NPC could be attributed to a combination with antioxidant activity and cholesterol clearance via the

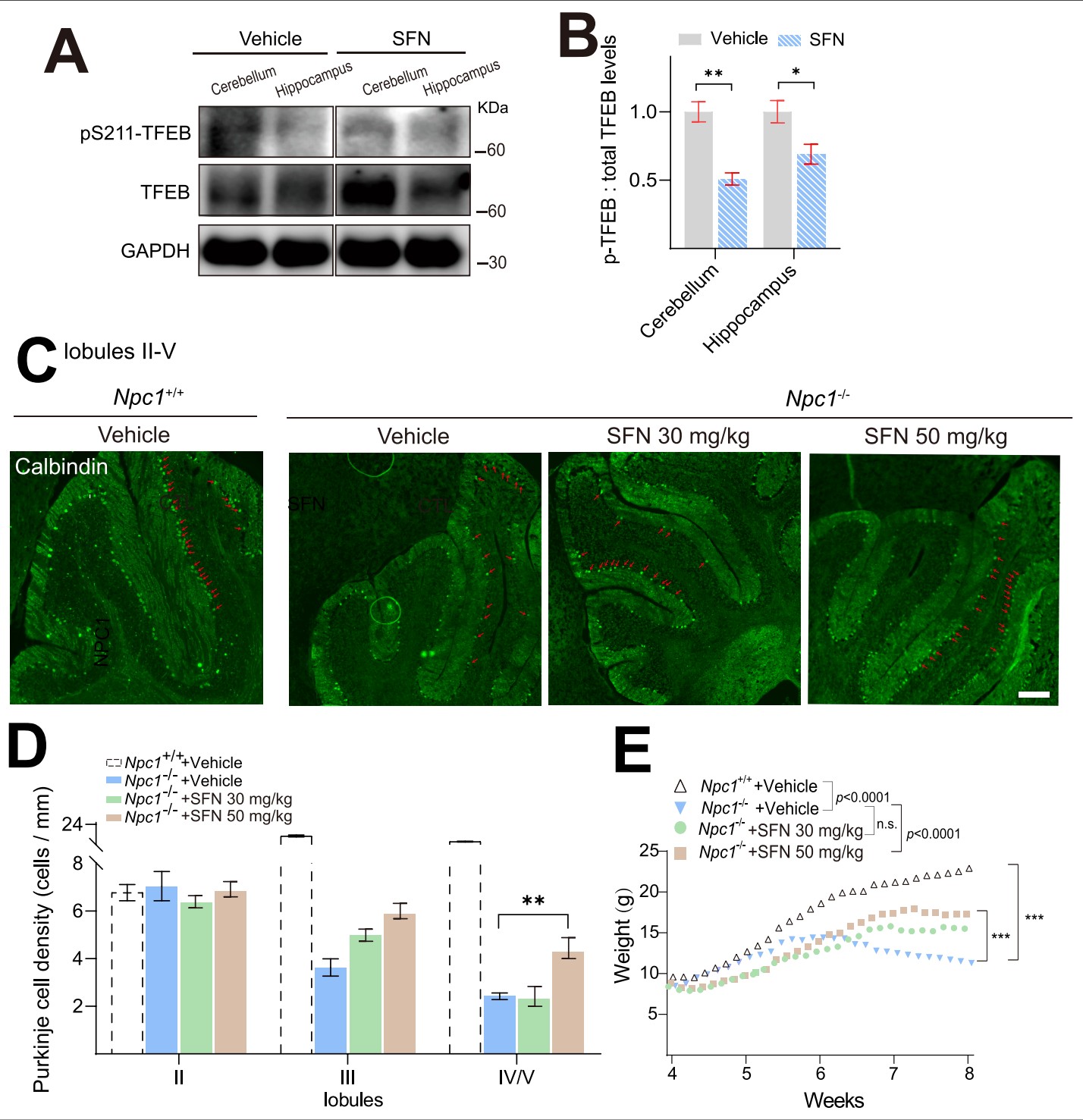

**Figure 6.** Sulforaphane (SFN) rescues the loss of Purkinje cells and body weight in NPC in vivo model mice. (**A**) SFN promoted TFEB dephosphorylation in mice brain. 4-week-old BALB/cJ mice were intraperitoneally (i.p.) injected with SFN (50 mg/kg) or vehicle for 12 h, and brain tissues including cerebellum and hippocampus were collected and subjected to detect pS211-TFEB and total TFEB levels by western blotting. (**B**) Quantification of the ratios of p-TFEB vs. total TFEB as shown in (**A**). (**C**) Cerebella from vehicle and SFN-treated NPC mice were analyzed at 8 weeks of age for calbindin by immunohistochemistry. SFN and vehicle were intraperitoneally injected daily in 4-week-old NPC mice for 4 weeks. Scale bar = 200 µm (n = 6 for each group). (**D**) Quantification of the number of Purkinje cells as indicated in the anterior lobules (II–V) as shown in (**C**). (**E**) Body weight was registered during the treatment. For all the panels, data are presented as mean ± s.e.m.; **p<0.01, ***p<0.001.

The online version of this article includes the following source data for figure 6:

*Figure 6 continued on next page*

*Figure 6 continued*
**Source data 1.** Original western blots for *Figure 6A*, indicating the relevant bands and treatments.
**Source data 2.** Original files for western blot analysis displayed in *Figure 6A*.

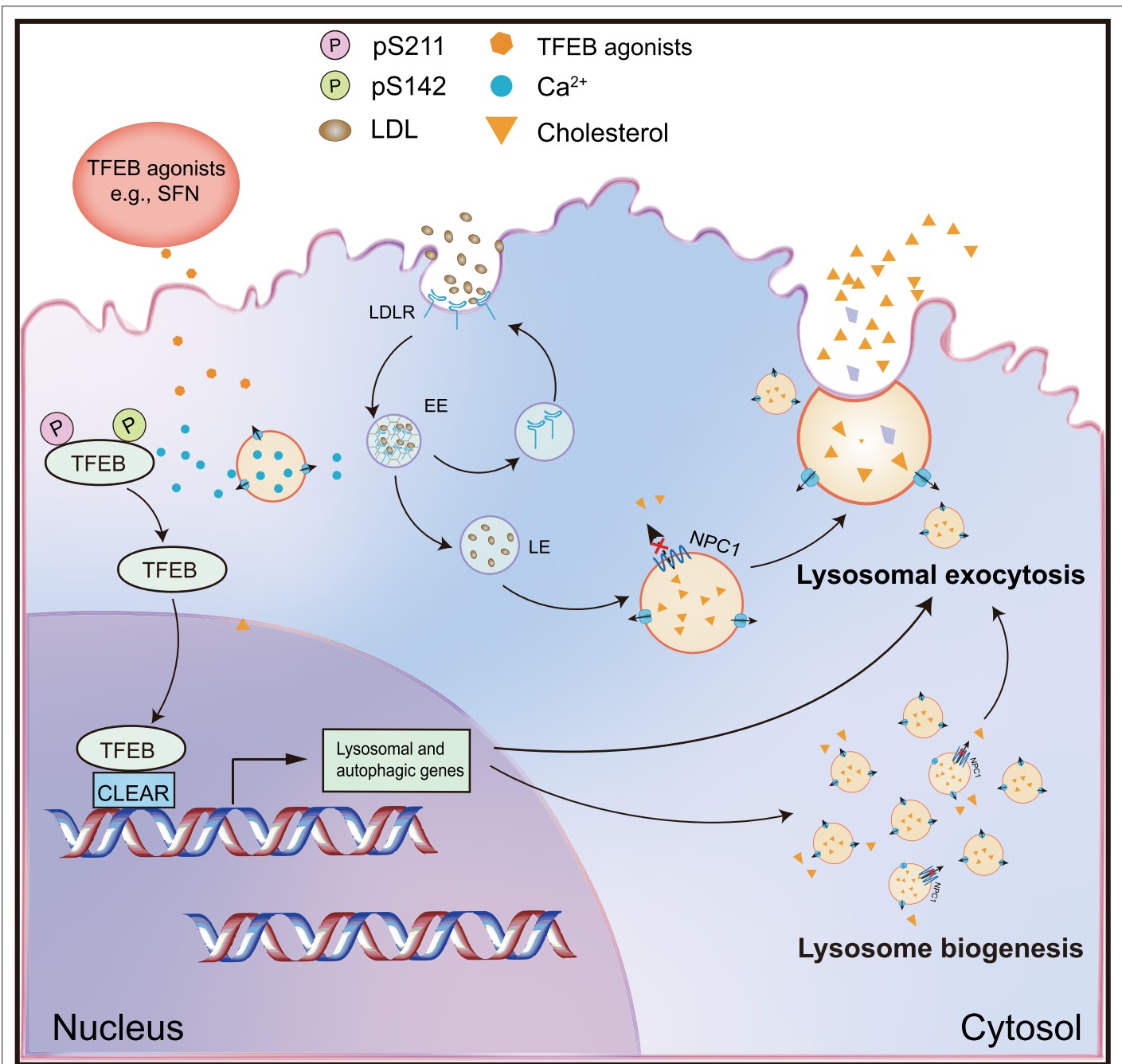

**Figure 7.** A working scheme to illustrate that small-molecule TFEB agonist promotes cholesterol clearance in the NPC model via TFEB-upregulated lysosomal exocytosis and biogenesis. Pharmacological or genetic activation/overexpression of TFEB dramatically ameliorates cholesterol accumulation in NPC1 cells. Small-molecule, BBB-permeable TFEB agonist sulforaphane (SFN) induces TFEB nuclear translocation by dephosphorylation of TFEB at S142 and S211 residues, promoting lysosomal biogenesis and exocytosis, resulting in mitigating lysosomal cholesterol levels.

lysosome-dependent, TFEB-mediated regulation. Therefore, pharmacological activation of TFEB may serve as a potential therapeutic strategy for NPC.

## Materials and methods

### Mammalian cell culture

HeLa cells were cultured in Dulbecco's Modified Eagle's Medium (DMEM; Thermo Fisher Scientific, 11195-065) supplemented with 10% fetal bovine serum (FBS; Thermo Fisher Scientific, 10091148). NPC1 patient-derived fibroblast cells (clone GM03123) and a healthy control (clone GM03440) were obtained from the Coriell Institute for Medical Research (NJ, USA). Human fibroblast cells were maintained in modified Eagle's medium (Thermo Fisher Scientific, 1964643) supplemented with 15% FBS, 2 mM glutamine (Thermo Fisher Scientific, 25030081), and 1% penicillin-streptomycin. $Npc1^{-/-}$ MEF cells were cultured in DMEM with 10% FBS, 1% penicillin-streptomycin, and 1% antibiotic-antimycotic (Thermo Fisher Scientific, 15240062). Macrophage cells were cultured in RPMI 1640 medium (Gibco, B122656) supplemented with 20% FBS. Unless otherwise indicated, all cell cultures were maintained at 37°C in a humidified 5% $CO_2$ incubator.

### Stable and CRISPR/Cas9 KO cell lines

GFP-TFEB stable HeLa cell line was kindly provided by Dr. Shawn Ferguson (Yale School of Medicine) (*Zhang et al., 2016*). *TFEB* CRISPR-Cas9 KO HeLa cells were generated and characterized as reported previously (*Li et al., 2021*).

### Plasmids/siRNA transfection

Plasmids, including mCherry-TFEB, mCherry-TFEB$^{S211A}$, and TFE3-GFP, were maintained in our laboratory as previously described (*Nezich et al., 2015*; *Li et al., 2021*). The siRNA sequences targeting human *TFEB* (5'-GAA AGG AGA CGA AGG UUC AAC AUC A-3') were purchased from Invitrogen. The siRNA sequences targeting human *NPC1* (5'-CAA UUG UGA UAG CAA UAU UTT-3') were chemically synthesized by GenePharma (Shanghai, China). HeLa cells were transfected with plasmids or siRNA using Lipofectamine 3000 reagent (Thermo Fisher Scientific, 2163785) or RNAi-Max reagent (Thermo Fisher Scientific, 13778150) in Opti-MEM (Thermo Fisher Scientific, 11058021), respectively. The efficiency of transfection was examined by western blotting or Q-PCR.

### Filipin staining

Cellular unesterified cholesterol was detected using a cell-based cholesterol assay kit (Abcam, ab133116). It provides a simple fluorometric method to detect the interaction of cholesterol and filipin III, which alters the filipin absorption and fluorescence spectra allowing visualization with a fluorescence microscope. Cells were cultured at $5 \times 10^2$ cells/well in black, clear-bottom 96-well plates and treated with compounds for the indicated conditions. After rinsing with PBS twice, cells were fixed with fixation solution for 10 min followed by a PBS rinse twice. The cells were then stained with filipin III solution for 1 h at room temperature in the dark. The images were then captured using an Olympus IX73/Zeiss microscope. Image analysis was conducted using the ImageJ software.

### Immunofluorescence and confocal imaging

For immunofluorescence detection of TFEB and Nrf2, cells were grown on coverslips, fixed with 4% PFA, permeabilized with 0.3% Triton X-100 (Solarbio, T8200), followed by blocking in 1% bovine serum albumin (BSA, Merck, B2064) in PBS for 1 h. Cells were then incubated with anti-TFEB (Cell Signaling Technology, 4240) or anti-Nrf2 antibody (Abcam, ab623521) at 4°C overnight. After 4 washes with PBS, coverslips were incubated with secondary antibodies for 1 h and counterstained with DAPI for 10 min. For LAMP1 immunostaining, cells were fixed with 100% methanol (–20°C) for 10 min and then blocked with 1% BSA in PBS for 1 h. The primary anti-LAMP1 (Abcam, ab24170) was used in this study. Finally, coverslips were mounted with Fluoromount-G (Southern Biotech, 0100-01) and ready for imaging.

### Western blotting

Cells were lysed with ice-cold RIPA buffer (Solarbio, R0010) in the presence of 1× protease inhibitor cocktail (Merck, P8340) and 1× phosphatase inhibitor cocktail (Abcam, GR304037-28) on ice for

20 min. Cells were then centrifuged and the supernatant was collected. The protein concentration of the supernatant was determined using BCA Protein Assay (Thermo Scientific, UA269551). Protein samples (20–40 μg) were then loaded and separated on SDS-polyacrylamide gradient gels (GenScript, M00654) followed by the transfer to polyvinylidene difluoride membranes (Merck, R7DA8778E). Western blot analysis was performed using primary antibodies against TFEB (1:500, Cell Signaling Technology, 4240 for cells, 1:1000, Bethyl Laboratories, A303-673A for mice tissue), pS122-TFEB (1:500, Cell Signaling Technology, 86843), pS142-TFEB (1:500, Millipore, 3321796), pS211-TFEB (1:500, Cell Signaling Technology, 37681), NPC1 (1:1000, Abcam, ab134113), LAMP1 (1:500, Abcam, ab24170), MTOR (Cell Signaling Technology, 2972), p-MTOR (Sigma-Aldrich, SAB4504476), p-RPS6KB1/S6K1 (Cell Signaling Technology, 9234), RPS6KB1/S6K1 (Cell Signaling Technology, 2708), NPC1 (1:1000, Santacruz, sc271335), and GAPDH (1:10,000, Sigma-Aldrich, G9545). Bound proteins were then detected with secondary antibodies against horseradish peroxidase-conjugated and enhanced chemiluminescence reagents (Thermo Fisher Scientific, 203-17071). The membranes were visualized using a Li-COR Biosciences Odyssey Fc system, and the band intensity was quantified using ImageJ software.

## LAMP1 surface labeling

Cells were pretreated with SFN as the indicated time. For the experiment of co-staining of DiO and surface LAMP1, cells were first incubated with 5 μM DiO (HY-D0969, MCE) at 37°C for 30 min and washed with PBS twice to remove the unbound dye. Nonpermeabilized cells were then labeled with anti-human LAMP1 antibody (1:500, DSHB, H4A3), which recognizes a luminal epitope, at 4°C for 1 h. Cells were then fixed in 2% paraformaldehyde for 30 min and incubated with Alexa Flour 488-conjugated secondary antibody at room temperature for 1 h. After PBS wash three times, cells were counterstained with DAPI for 10 min and images were captured using Zeiss confocal microscope.

## Lysosomal pH imaging

To measure lysosomal luminal pH, live cells were treated with chemicals as the indicated condition, followed by incubation with 50 nM LysoTracker Red DND-99 (Thermo Scientific, L7528) for 15–30 min. Cells were then washed with PBS for three times. Images were captured using an Olympus/Zeiss microscope. Fluorescence intensities were quantified using the Image J software.

Lysosomal luminal acidity was also evaluated with the fluorescence ratio between a pH-sensitive dye pHrodo Green conjugated dextran-10 kd (P35368, Invitrogen) and a pH-insensitive dye CF555 conjugated dextran-10 kd (80112, Biotium). Briefly, cells were seeded on coverslips and loaded with 20 μg/ml of each dextran overnight. Cells were then chased in medium without dye for 3 h before imaging. The HEPES buffered DMEM medium without phenol red (21063029, Gibco) was used as the imaging solution to eliminate the short-time starvation side effects during the imaging process. Cells were washed with the imaging solution and imaged using an inverted microscope (Olympus IX81). The fluorescence emission excited at 488 nm and 561 nm wavelengths were acquired with an EM-CCD camera. The fluorescence intensity of pHrodo Green and CF555 were quantified using ImageJ.

## RNA extraction and RT-QPCR

Total RNA was extracted using TRIzol according to the manufacturer's protocol (Thermo Scientific, 191002). cDNA was generated with 100–500 ng of total RNA using GoScript Reverse Transcription System (Promega, 0000316057). Q-PCR was performed using SYBR Green (TOYOBO, 563700) in CFX Connect Optics (Bio-Rad). The changes in the mRNA expression of target genes were normalized to that of the housekeeping gene HPRT. The primer sequences used in this study are listed as follows:

*HPRT*: For 5'-tggcgtcgtgattagtgatg-3', Rev 5'-CTGTTCTCGTCCAGCAGACACT-3'
*LAMP1*: For 5'-CGTGTCACGAAGGCGTTTTCAG-3', Rev 5'-CTGTTCTCGTCCAGCAGACACT-3'
*ULK1*: For 5'-TCATCTTCAGCCACGCTGT-3', Rev 5'-CACGGTGCTGGAACATCTC-3'
*SQSTM1*: For 5'-CTGGGACTGAGAAGGCTCAC-3'; Rev 5'-GCAGCTGATGGTTTGGAAAT-3'
*ATG5*: For 5'-TGCGGTTGAGGCTCACTTTATGTC-3'; Rev 5'-GTCCCATCCAGAGCTGCTTGTG-3'
*mGAPDH*: For 5'-TGA ATA CGG CTA CAG CA-3'; Rev 5'-AGG CCC CTC CTG TTATTA TG-3'
*mSQSTM1*: For 5'-AGGAGGAGACGATGACTGGACAC-3'; Rev 5'-TTGGTCTGTAGGAGCC TGGTGAG-3'

*mLC3*: For 5′-CAAGCCTTCTTCCTCCTGGTGAATG-3′; Rev 5′-CCATTGCTGTCCCGAATGTC
TCC-3′

*mCTSF*: 5′-ACGCCTATGCAGCCATAAAG-3′; Rev 5′-CTTTTGCCATCTGTGCTGAG-3′

## Measurement of NAGase and ACP activity

Activities of NAGase and ACP enzymes were measured using microplate assay kits (NAGase, absin, abs580171; ACP, Solarbio, BC2135) respectively. Following the manufacturer's instructions, cells were seeded in 6-well plates and treated with chemicals in FBS-free medium as the indicated condition. Aliquot of supernatant medium was collected and put on ice for extracellular (medium) enzyme activity detection. Cells were collected and 100 µl of assay buffer was added. The cell suspension was then sonicated and centrifuged for 8000 × *g* for 10 min, and supernatant was collected for cellular enzyme activity detection. NAGase activity was measured in a 96-well microtiter plate containing 25 µl of sample (medium or cell lysates) and 25 µl of substrate in each well, which was mixed and incubated at 37°C for 20 min, and then 50 µl of stop solution was added to stop the reaction. ACP activity was measured in a microtiter plate containing 20 µl of sample, 40 µl of Reagent I and 40 µl of Reagent II. The plate was incubated at 37°C for 15 min and then 120 µl of Reagent III was added. Finally, the absorbance of NAGase/ACP was recorded at 405 nm/510 nm using a FlexStation 3 Multi-Mode microplate reader (Molecular Devices), respectively. NAGase/ACP activity released to medium (%)=Enzyme activity in medium/total enzyme activity (medium + cell lysates).

## Detection of cholesterol content

The levels of total and free cholesterol released in the medium were measured using a colorimetric cholesterol/cholesteryl ester detection kit (Abcam, ab102515) according to the manufacturer's instructions. Briefly, aliquot of supernatant medium was collected and air-dried at 50°C. The dried mixture was dissolved in cholesterol assay buffer. Aliquot of samples was mixed with cholesterol assay buffer, substrate, and cholesterol enzyme with or without cholesterol esterase and incubated at 37°C for 30 min. The absorbance was measured at 450 nm using a FlexStation 3 Multi-Mode microplate reader (Molecular Devices). The amount of cholesterol was calculated by standard curve and normalized with cellular protein content.

## Cell viability assay

Cell viability was assessed by MTT assay (Merck, M2128-5G). A modified MTT assay was applied to measure the cell viability. Briefly, approximately $10^4$ cells were seeded in 96-well plates and exposed to SFN as required. Cells were then incubated in fresh medium containing 0.2 mg/ml MTT solution for 4 h. The supernatant was removed, and 100 µl of acidified DMSO (0.04 M HCl/DMSO) was added to dissolve the precipitation at 37°C for 10 min. The absorbance of dissolved solution was measured at 490 nm with Flexstation 3 (Molecular Devices). Cell mortality (%) was calculated by $(OD^{Control}-OD^{Sample})/OD^{Control} \times 100$.

## Animals

*Npc1$^{-/-}$* mice (BALB/cNctr-Npc1 m1N/J, also known as NPC1$^{NIH}$) were purchased from The Jackson Laboratory (USA). All the experimental procedures were conducted in accordance with the Guide for the Care and Use of Laboratory Animals in the Zhejiang University of Technology (Hangzhou, China) and conformed to the National Institutes of Health Guide for Care and Use of Laboratory Animals. Unless stated otherwise, mice were fed with free access to water and standard diet under specific pathogen-free conditions. Genotypes were identified using a PCR-based screening (*Amigo et al., 2002*).

## Histological analysis

Mice perfusion was performed with PBS. Then, mice cerebellums were post-fixed overnight at 4°C and then placed in serial dilutions of sucrose (10–30%) in PBS at 4°C overnight, respectively. Then cerebellums were cut in 5-µM-thick sagittal sections by cryostat at (Leika) at –20°C. Permeabilized slices with 0.1% Triton X-100 were blocked in 1% BSA in PBS for 1 h. Slices were incubated with anti-calbindin (Abcam, 108404) overnight at 4°C, followed by incubation with the secondary antibody conjugated with Alexa Fluor 488 for 2 h. The slices were then washed with PBS three times and

incubated with DAPI in the dark for 10 min. All the images were captured with an Olympus or Zeiss confocal microscope.

## Isolation of primary MEF and macrophage cells

MEF cells were prepared from neonatal mice, which were euthanized by $CO_2$. Skin of neonatal mice was cut with scissors and gently clipped and added to 0.25% trypsin-EDTA in a 37°C incubator for 20 min. The cell suspension was then transferred to a 50 ml tube, and 10 ml of DMEM media was added to inactivate the trypsin reaction for 5 min. The supernatant was transferred to a 60 mm dish and kept in the incubator at 37°C for 3 h, then the medium was replaced by DMEM with 10% FBS, 1% penicillin/streptomycin until confluency (2–4 days). Primary macrophages were established from hindlimb femurs and tibias of newborn mice. The marrow cells were flushed from the bones with PBS and centrifuged. Cells were then resuspended in RPMI 1640 medium (Gibco, B122656) supplemented with 20% FBS. Cells were then seeded in culture dishes coated with 2% gelatin and allowed to adhere for 2 h at 37°C.

## Reagents

The chemicals used in this study include SFN (Sigma-Aldrich, S4441), DMSO (Sigma-Aldrich, D2660), filipin III (Cayman Chemical, 70440), U18666A (MedChemExpress, HY-107433), Triton X-100 (MCE, HY-Y1883A), NAC (Sigma-Aldrich, A7250), FK506 (Sigma-Aldrich, F4679), CsA (Solarbio, C8780), and BAPTA-AM (Thermo Scientific, 1824047).

## Data analysis

Data are presented as mean ± s.e.m. from at least three independent experiments. Statistical comparisons were performed with ANOVA or Student's *t*-test with paired or unpaired wherever appropriate. A p-value<0.05 was considered statistically significant.

## Acknowledgements

This work was supported by an NSFC grant (31600823 to DL). Additional support was provided by Calygene Biotechnology Inc (XT [2016]008@) and Lysoway Therapeutics Inc (KYY-HX-20210129). The funders had no role in the study design, data collection and analysis, decision to publish or preparation of the manuscript. We are grateful to Dr. Shawn M Ferguson for the GFP-TFEB stable cells and Dr. Haoxing Xu for the TFEB KO cells.

---

## Additional information

### Funding

| Funder | Grant reference number | Author |
|---|---|---|
| National Natural Science Foundation of China | 31600823 | Dan Li |
| Calygene Biotechnology Inc | XT [2016]008@ | Dan Li |
| Lysoway Therapeutics, Inc | KYY-HX-20210129 | Dan Li |

The funders had no role in study design, data collection and interpretation, or the decision to submit the work for publication.

### Author contributions

Kaili Du, Data curation, Formal analysis, Investigation, Methodology, Writing – original draft; Hongyu Chen, Data curation, Validation, Investigation, Methodology, Writing – review and editing; Zhaonan Pan, Yu Luo, Validation, Investigation, Methodology; Mengli Zhao, Data curation, Investigation, Methodology, Writing – review and editing; Shixue Cheng, Investigation, Methodology; Wenhe Zhang, Formal analysis, Writing – review and editing; Dan Li, Conceptualization, Data curation, Supervision,

Funding acquisition, Validation, Investigation, Methodology, Writing – original draft, Project administration, Writing – review and editing

### Author ORCIDs
Dan Li https://orcid.org/0000-0002-7316-5151

### Ethics
This study was performed in strict accordance with the recommendations in the Guide for the Care and Use of Laboratory Animals of the Zhejiang University of Technology. All of the animals were handled according to approved institutional animal care and use committee (IACUC) protocols (20210407016) of the Zhejiang University of Technology. The protocol was approved by the Committee on the Ethics of Animal Experiments of the University of Zhejiang University of Technology (Permit Number: SYXK 2022-0007). All surgery was performed under sodium pentobarbital anesthesia, and every effort was made to minimize suffering.

Reviewer #2 (Public review): https://doi.org/10.7554/eLife.103137.3.sa1
Author response https://doi.org/10.7554/eLife.103137.3.sa2

## Additional files

### Supplementary files
MDAR checklist

Source data 1. All original prism graphs in the main figures.

Source data 2. All original prism graphs in the figure supplements.

### Data availability
All raw imaging data have been deposited in BioImage Archive under accession code S-BIAD1694.

The following dataset was generated:

| Author(s) | Year | Dataset title | Dataset URL | Database and Identifier |
|---|---|---|---|---|
| Du K, Chen H, Pan Z, Zhao M, Cheng S, Luo Y, Zhang W, Li D | 2025 | Small-molecule Activation of TFEB Alleviates Niemann-Pick Disease Type C via Promoting Lysosomal Exocytosis and Biogenesis | https://doi.org/10.6019/S-BIAD1694 | BioImages, 10.6019/S-BIAD1694 |

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
