## [Editor Report · eLife Assessment]

This study reports that activation of TFEB promotes lysosomal exocytosis and clearance of cholesterol from lysosomes, the strength of evidence for which is **convincing** with appropriate and validated methodology in line with current state-of-the-art. The significance of the findings is **important** in the context of Niemann–Pick disease type C as well as other subfields.

---

## [Referee Report · Reviewer #2 (Public review)]

Summary:

This study presents an important finding that the activation of TFEB by sulforaphane (SFN) could promote lysosomal exocytosis and biogenesis in NPC, suggesting a potential mechanism by SFN for the removal of cholesterol accumulation, which may contribute to the development of new therapeutic approaches for NPC treatment.

Strengths:

The cell-based assays are convincing, utilizing appropriate and validated methodologies to support the conclusion that SFN facilitates the removal of lysosomal cholesterol via TFEB activation.

Comments on revisions:

The authors have addressed most of my questions. I have only one minor technical point to emphasize, which does not affect the overall strength of the evidence for this project.

The pKa values of pHrodo Green (P35368, pKa=6.757) and pHrodo Red-Dex (P10361, pKa=6.816) are very similar. Prof. Xu's article, cited in the response letter (Hu, Li et al. 2022), is an excellent example of lysosomal pH measurement. He used LysoTracker Red DND-99 for a rough estimation of lysosomal acidity, and for accurate monitoring of lysosomal pH, he employed the ratiometric OG488-dex (pKa 4.6).

---

## [Author Response]

The following is the authors’ response to the original reviews.

Although the reviewers found our work interesting, they raised several important concerns about our study. To address these concerns, mostly we performed new experiments. The most important changes are highlighted in the summary paragraphs.

First, in response to Reviewer 1’s suggestions, we have conducted the SFN experiments systematically, e.g., we further confirmed the mechanism of SFN-activated TFEB in HeLa NPC1 cells with new experiments including: the effect of BAPTA-AM (a calcium chelator), FK506+CsA (calcineurin inhibitors) and NAC (ROS scavenger) on SFN-induced TFEB-nuclear translocation in HeLa NPC1 cells (New Fig. S3). The effect of SFN on NPC1 expression (New Fig. S5). Particularly, we examined the colocalization of DiO (a PM marker) staining and surface LAMP1 staining in HeLa NPC1 cells under SFN treatment to confirm the PM exocytosis. In main text and figure legends, accuracy of sentence is thoroughly checked and defined. Hence, we have significantly improved the presentation and clarity in the revision.

Second, in response to Reviewer 2’s suggestions, we have performed additional experiments to demonstrate that the role of TFEB in SFN-evoked the lysosomal exocytosis by using TFEB-KO cells (New Fig. S7B). In TFEB KO cells, this increase of surface LAMP1 signal by SFN treatment was significantly reduced, suggestive of SFN-induced exocytosis in a TFEB-dependent manner. We also investigated the effect of U18666A on CF555-dextran endocytosis. By examining the localization of CF-dex and Lamp1, we found that CF555 is present in the lysosome with U18666A treatment (Fig for reviewers only A,B), suggesting that NPC1 deficiency/U18666A treatment has no effect on CF-dex endocytosis.

Third, in response to Reviewer 3’s suggestions, we have performed experiments in addition to response to other reviewers’ suggestion ie. the cytotoxicity of the concentration of SFN used in this study in various cell lines (New Fig.S10).

In addition, according to the reviewers’ suggestions, we made clarifications and corrections wherever appropriate in the manuscript.

**Reviewer #1 (Public review):**
Summary:The authors are trying to determine if SFN treatment results in dephosphorylation of TFEB, subsequent activation of autophagy-related genes, exocytosis of lysosomes, and reduction in lysosomal cholesterol levels in models of NPC disease.Strengths:(1) Clear evidence that SFN results in translocation of TFEB to the nucleus.(2) In vivo data demonstrating that SFN can rescue Purkinje neuron number and weight in NPC1^-/-^ animals.

Thank you for the support!

Weaknesses:(1) Lack of molecular details regarding how SFN results in dephosphorylation of TFEB leading to activation of the aforementioned pathways. Currently, datasets represent correlations.

Thank you for raising this critical point! The reviewer is right that in this manuscript we did not talk too much about the molecular mechanism of SFN-evoked TFEB activation. Because in our previous study (Li, Shao et al. 2021), we explored the mechanism of SFN-induced TFEB activation. We show that SFN-evoked TFEB activation via a ROS-Ca^2+^-calcineurin dependent but MTOR -independent pathway (Li, Shao et al. 2021). In the current manuscript, we cited this paper, but did not talk the details of the mechanism, which obviously confused the reviewers. Therefore, in the revision manuscript we added more details of the molecular mechanism of SFN-activated TFEB. Also, we further confirmed this mechanism in HeLa NPC1 cells with new experiments including: the effect of BAPTA-AM (a calcium chelator), FK506+CsA (calcineurin inhibitors) and NAC (ROS scavenger) on SFN-induced TFEB-nuclear translocation in NPC cells (New Fig.S3).

(2) Based on the manuscript narrative, discussion, and data it is unclear exactly how steady-state cholesterol would change in models of NPC disease following SFN treatment. Yes, there is good evidence that lysosomal flux to (and presumably across) the plasma membrane increases with SFN. However, lysosomal biogenesis genes also seem to be increasing. Given that NPC inhibition, NPC1 knockout, or NPC1 disease mutations are constitutively present and the cell models of NPC disease contain lysosomes (even with SFN) how could a simple increase in lysosomal flux decrease cholesterol levels? It would seem important to quantify the number of lysosomes per cell in each condition to begin to disentangle differences in steady state number of lysosomes, number of new lysosomes, and number of lysosomes being exocytosed.

Thank you for this constructive comment. From our data, in NPC1 cells SFN reduced the cholesterol levels by inducing lysosomal exocytosis and increasing lysosomal biogenesis. We understand the reviewer’s point that it would be really helpful to differentiate the exact three states of original number of lysosomes, number of new lysosomes, and number of lysosomes being exocytosis. Unfortunately, due to the technique limitation, so far seems there is no appropriate method that could clearly differentiate the lysosomes exactly come from which state. In the future, hopefully we will have technique to explore this mechanism.

(3) Lack of evidence supporting the authors' premise that "SFN could be a good therapeutic candidate for neuropathology in NPC disease".

Suggestion was taken! We removed this sentence. Thanks!

**Reviewer #2 (Public review):**
(4) The in vivo experiments demonstrate the therapeutic potential of SFN for NPC. A clear dose response analysis would further strengthen the proposed therapeutic mechanism of SFN.

Thank you for this constructive suggestion. We examined the effect of two doses of SFN30 and 50mg/kg on NPC mice. As shown in Fig.6, SFN (50mg/kg), but not 30mg/kg prevents a degree of Purkinje cell loss in the lobule IV/V of cerebellum, suggesting a dose-correlated preventive effect of SFN. In the future study, we will continue optimizing the dosage form and amount of SFN and do a dose-responsive analysis.

(5) Additional data supporting the activation of TFEB by SFN for cholesterol clearance in vivo would strengthen the overall impact of the study.

Thank the reviewer for this constructive comment. We have detected a significant decrease of pS211-TFEB protein in brain tissues of NPC mice upon SFN treatment compared to vehicle, suggesting that SFN activates TFEB in brain tissue for the first time. It is worth to further examine the lysosomal cholesterol levels in brain tissues to show the direct effect of SFN. However, in our hands and in the literatures Filipin seems not suitable for detecting lysosomal cholesterol accumulation in brain tissue. So far there isn’t a good method to directly measure lysosomal cholesterol in tissue.

(6) In Figure 4, the authors demonstrate increased lysosomal exocytosis and biogenesis by SFN in NPC cells. Including a TFEB-KO/KD in this assay would provide additional validation of whether these effects are TFEB-dependent.

Great suggestion! We investigated the role of TFEB in SFN-evoked the lysosomal exocytosis by using TFEB-KO cells. As shown in New Suppl. Fig. 7B, in TFEB KO cells, this increase of surface LAMP1 signal by SFN (15 μM, 12 h) treatment was significantly reduced, suggestive of SFN induced exocytosis in a TFEB-dependent manner.

(7) For lysosomal pH measurement, the combination of pHrodo-dex and CF-dex enables ratiometric pH measurement. However, the pKa of pHrodo red-dex (according to Invitrogen) is ~6.8, while lysosomal pH is typically around 4.7. This discrepancy may account for the lack of observed lysosomal pH changes between WT and U18666A-treated cells. Notably, previous studies (PMID: 28742019) have reported an increase in lysosomal pH in U18666A-treated cells.

We understand the reviewer’s point. But as stated in the methods and main text, we used pHrodo Green-Dextran (P35368, Invitrogen), rather than pHrodo Red-dextran. According to the product information from Invitrogen, pHrodo Green-dex conjugates are non-fluorescent at neural pH, but fluorescence bright green at acidic pH around 4, such as those in endosomes and lysosomes. Therefore, pHrodo Green-dex is suitable to monitor the acidity of lysosome (Hu, Li et al. 2022). We also used LysoTracker Red DND-99 (Thermo Scien fic, L7528) to measure lysosomal pH (Fig. 4G, H), which is consistent with results from pHrodo Green/CF measurement.

The reviewer mentioned that previous studies have reported an increase in lysosomal pH in U18666Atreated cells. We understood this concern. But in our hands, from our data with two lysosomal pH sensors, we have not detected lysosomal pH change in U18666A-treated NPC1 cell models.

(7) The authors are also encouraged to perform colocalization studies between CF-dex and a lysosomal marker, as some researchers may be concerned that NPC1 deficiency could reduce or block the trafficking of dextran along endocytosis.

Thank you for raising this important point and suggestion was taken! We investigated the effect of NPC1 deficiency on CF555-dextran trafficking into lysosome by examining the localization of CF-dex and Lamp1. To clearly define whether CF555-dex is present in the lysosome, we first used apilimod to enlarge lysosomes and then examined the relative posi on of CF555-dex and lamp1. As shown in Author response image 1A,B, in HeLa cells treated with U18666A, CF555 signals (red) clearly present inside lysosome (LAMP1 labelled lysosomal membrane, green signal), suggesting that CF555dex endocytosis is not affected by NPC1 deficiency (U18666A treatment).

**Author response image 1. sa2fig1:** The effect of NPC1 deficiency on CF555 endocytosis. HeLa cells were transiently transfected with LAMP1-GFP plasmid for 24 h. Cells were then treated with apilimod (100 nM) for 2 h to enlarge the lysosomes, and followed by co- treatment of U18666A (2.5 μM, 24 h) and CF555 (12 h). (A) Each panel shows fluorescence images taken by confocal microscopes. (B) Each panel shows the fluorescence intensity of a line scan (white line) through the double labeled object indicated by the white arrow. Scale bar, 20 μm or 2 μm (for zoom-in images).

(9) In vivo data supporting the activation of TFEB by SFN for cholesterol clearance would significantly enhance the impact of the study. For example, measuring whole-animal or brain cholesterol levels would provide stronger evidence of SFN's therapeutic potential.

We really appreciate the reviewer’s comments. Please see response to point #5.

**Reviewer #3 (Public review):**
(10) The manuscript is extremely hard to read due to the writing; it needs careful editing for grammar and English.

Sorry for the defects in the writing and grammar. We had thoroughly checked grammar and polished the English to improve the manuscript.

(11) There are a number of important technical issues that need to be addressed.

We will address the technical issues mentioned in the following ques ons.

(12) The TFEB influence on filipin staining in Figure 1A is somewhat subtle. In the mCherry alone panels there is a transfected cell with no filipin staining and the mCherry-TFEBS211A cells still show some filipin staining.

Thank you for raising this point. The reviewer is right that not all the mCherry alone cells with the same level of filipin signal and not all mCherry-TFEBS211 transfected cells show completely no filipin signal. The statistical results were from randomly selected cells from 3 independent experiments. To avoid the confusion, we have included more cells in the statistical analysis to cover all the conditions as shown in the new Fig. 1B. Hopefully this helps to clarify the confusion.

(13) Figure 1C is impressive for the upregulation of filipin with U18666A treatment. However, SFN is used at 15 microM. This must be hitting multiple pathways. Vauzour et al (PMID: 20166144) use SFN at 10 nM to 1microM. Other manuscripts use it in the low microM range. The authors should repeat at least some key experiments using SFN at a range of concentrations from perhaps 100 nM to 5 microM. The use of 15 microM throughout is an overall concern.

The reason that we use this concentration of SFN is based on our previous study (Li, Shao et al. 2021). We had shown that SFN (10–15 μM, 2–9 h) induces robust TFEB nuclear translocation in a dose- and time-dependent manner in HeLa cells as well as in other human cell lines without cytotoxicity (Li, Shao et al. 2021). Also, tissue concentrations of SFN can reach 3–30 μM upon broccoli consumption (Hu, Khor et al. 2006), so we used low micromolar concentrations of SFN (15 μM) in our study. Moreover, we further confirmed that SFN (15 μM) induces TFEB nuclear translocation in HeLa NPC1 cells (Fig. 1F, G Fig. 2B, G) and this concentration of SFN has no cytotoxicity (New Fig.S10).

**Recommendations for the authors:**

**Reviewer #1 (Recommendations for the authors):**
The following comments are designed to improve and focus the authors' work.(14) Related to data in Figure 1. The mechanism through which TFEB can reduce Filipin in U18 conditions is unclear. Inhibi on of NPC1 results in hyperactivation of mTOR through cholesterol transport at ER-Lysosome contacts (see Zoncu group publications). If mTORC is hyperac ve in NPC disease models, TFEB would be expected to remain cytoplasmic and not enter the nucleus as the representative image in Figure 1A demonstrates.

In our previous study (Li, Shao et al. 2021), we have shown that SFN induces TFEB nuclear translocation in a mTOR-independent manner (Li, Shao et al. 2021). Consistent with this result, in this study we confirmed that SFN-induced TFEB nuclear translocation is mTor-independent in NPC1 cells (Now Fig. S4A, B). Thus, SFN induced TFEB nuclear translocation in various NPC cells (Fig. 1F, G, Fig. 2B, G). Please also see the discussion about the mechanism of SFN in response to point #1.

(15) Therefore, how does overexpression of TFEB, which remains in the cytoplasm, result in a decreased filipin signal? Similar ques ons relate to Figure 1C-H.

Medina et. al (Medina, Fraldi et al. 2011) show that TFEB overexpression (not activation, so overexpressed TFEB is in the cytoplasm) increases the pool of lysosomes in the proximity of the plasma membrane and promotes their fusion with PM by raising intracellular Ca^2+^ levels through lysosomal Ca^2+^ channel MCOLN1, leading to increased lysosomal exocytosis. Hence, TFEB overexpression only (TFEB is not activated) could reduce filipin signal via increasing lysosomal exocytosis. And with TFEB agonist treatment such as TFEB could further boost this increase.

(16) It would seem appropriate to measure the NPC1 and NPC2 proteins using western blot to ensure that SFN-dependent clearance of cholesterol is not due to enhanced expression of the native protein in U18-treated cells or enhanced folding of the protein in patient fibroblasts.

Thank you for this constructive comment! Because NPC1 gene mutation takes about 95% of NPC cases and NPC2 mutation takes about 5% of NPC cases. And in this study we focused on NPC1 deficiency cases. Thus, we measured the effect of SFN on the expression of NPC1 in human NPC1-patient fibroblasts. Western blot analysis showed that SFN (15 μM, 24 h) treatment did not affect NPC1 expression in human NPC1-patient fibroblasts (new Fig. S5).

(17) Related to data in Figures 1C-E. Controls are missing related to the effect SFN has on steady-state cholesterol levels. This may be insightful in providing information on the mode of action of this compound.

Suggestion was taken! We have supplemented the control- SFN only in new Fig. 1C-E.

(18) The mechanism that links SFN to TFEB-dependent translocation is suggested to involve calcineur independent dephosphorylation of TFEB. However, no data is provided. It would seem important to iden fy the mechanism(s) through which SFN positively regulates TFEB location. This would shift the manuscript and its model from correlations to causation. Experiments involving calcineurin inhibitors, or agonists of TRPML1 that have been reported as being a key source of Ca^2+^ for calcineurin activation, may provide molecular insight.

Please see the paragraph in response to point #1.

(19) Related to Figure 4. Using a plasma membrane counterstain to quantify plasma membrane LAMP1 would increase the rigor of the analysis.

Great idea! We examined the colocalization of DiO (a PM marker) staining and LAMP1 staining in HeLa NPC1 cells under SFN treatment. As shown in new Fig.4A, surface LAMP1 signal(red) colocalized with DiO (green), a PM marker.

(20) Related to Figure 5. How do the authors explain the kinetic disparity between SFN treatment for 24 vs 72 hrs? IF TFEB is activated and promoting lysosomal biogenesis and increased lysosomal flux across the PM, why does cholesterol accumulation lag? Perhaps related to this point. Are other cholesterol metabolizing enzymes that may have altered activity in NPC sensitive to SFN? A similar comment applies to the Sterol regulatory element binding protein pathway, which has been shown to be activated in models of NPC disease.

We understand the reviewer’s point. As shown in Fig. 5C, D, in NPC1^-/-^ MEF cells, SFN treatment for 24 h showed relative weaker cholesterol clearance compared to the effects in human cells (Fig.1C, D, Fig.2.E, I). Thus, we explored a longer treatment of SFN for 72 h (fresh SFN in medium was added every 24 h), and 72h treatment of SFN exhibited substantial cholesterol reduction (Fig. 5C, D). This different effect could be attributed to the continuous action of SFN, which could prolong the exocytosis, leading to more effective cholesterol clearance. As shown in the DMSO-treated MEF cells, the cholesterol levels are similar in both 24 and 72 h, thus 24 h U18666A treatment has reached the upper limit of the accumulated cholesterol, longer treatment me would not change the cholesterol levels. Thus, cholesterol accumulation has no lag.

We did not investigate whether SFN regulates other cholesterol metabolizing enzymes or sterol regulatory element binding proteins although we cannot rule out this possibility. In this study we mainly focus on the cholesterol clearance effect by SFN via TFEB-mediated pathways. From our data, TFEB KO could significantly diminish SFN-evoked cholesterol clearance. Hence, the effect of other cholesterol metabolizing enzymes or sterol regulatory element binding proteins maybe not as important as TFEB, thus out of scope of this study. In the future, we may explore the involvement of possible other pathways on SFN’s effects.

(21) Related to Figure 7. The western blots for pS211-TFEB are poor. It's suggested that whole blots are shown to increase rigor.

Thank you for the comments. We have represented the blots with more spare space to increase the rigor.

(22) Data demonstrating the ability of SFN to improve Purkinje cell survival are exci ng and pair well with the weight analysis, however, to address the overall goal of determining if "SFN could be a good therapeutic candidate for neuropathology in NPC disease" survival analysis should be tested as well.

Please see the paragraph in response to point #3.

Minor(23) Throughout the manuscript many different Fonts and font sizes are used. This is very jarring to readers. It is suggested that a more uniform approach is taken to presenting these nice datasets.

We are so sorry and apologize for these oversights. We have thoroughly checked all the manuscript to make sure that Fonts and sizes of font are synchronized.

(24) Related to data presentation. In general, there is a lack of alignment and organization of the figures.

So sorry about this. We have reorganized the figures to get them better aligned.

(25) Line 149, SFN is missing.

Corrected!

**Reviewer #3 (Recommendations for the authors):**
(26) In Figure 3 the authors should use multiple single siRNAs or perform a functional rescue to determine specificity.

We understand the reviewer’s point. We did design several siRNAs and the efficiency of these siRNAs were validated. Finally, we decide use this siRNA whose knockdown efficiency is best in the study and the specificity of the siTFEB has been validated by Western blot as shown in Fig. 3A. Furthermore, we used TFEB knockout cells constructed by CRISPR/Cas9 to further examine the role of TFEB in SFN-induced cholesterol clearance (Fig. 3D). Consistently with the results in the siTFEB-transfected HeLa NPC1 cells (Fig. 3B, C), SFN failed to diminish cholesterol in HeLa TFEB KO cells. The result from TFEB KO cells is even convincing than siRNA experiment. We also performed a functional rescue of re-expressing TFEB in TFEB KO cells, in which SFN-induced cholesterol clearance was restored (Fig. 3E, F). Collectively, these data indicate that TFEB is required for lysosomal cholesterol reduction upon SFN treatment. Thus, we did not repeat this rescue experiment in the siTFEB-transfected HeLa NPC1 cells.

(27) The label for 3D is missing.

Corrected! Thanks!

(28) Figure 4, although the authors use an an body against the luminal domain of LAMP1 there could s ll be some permeabilization. A marker of the plasma membrane would be helpful.

Please see the response to point #19.

(29) Figure 4, cholesterol in the media because of lysosome exocytosis. This is where the high concentration of SFN is of concern. Is there any cell death that could explain the result? The authors should test for cell death with the SFN treatment.

Thank you for raising this important point! We have measured the cytotoxicity of SFN of the concentrations used in this study in various cell lines (New Fig.S10). Please also see the paragraph in response to point #13.

(30) The blot in Figure 6A is unclear. It is very hard to see any change in pS211-TFEB levels, and, the blurry signal is the detection of phospho-TFEB is uncertain.

Please see the summary paragraph in response to point #21.

References:

Hu, M. Q., P. Li, C. Wang, X. H. Feng, Q. Geng, W. Chen, M. Marthi, W. L. Zhang, C. L. Gao, W. Reid, J. Swanson, W. L. Du, R. Hume and H. X. Xu (2022). "Parkinson's disease-risk protein TMEM175 is a proton-activated proton channel in lysosomes." Cell 185(13): 2292-+.

Hu, R., T. O. Khor, G. Shen, W. S. Jeong, V. Hebbar, C. Chen, C. Xu, B. Reddy, K. Chada and A. N. Kong (2006). "Cancer chemoprevention of intestinal polyposis in ApcMin/+ mice by sulforaphane, a natural product derived from cruciferous vegetable." Carcinogenesis 27(10): 2038-2046.

Li, D., R. Shao, N. Wang, N. Zhou, K. Du, J. Shi, Y. Wang, Z. Zhao, X. Ye, X. Zhang and H. Xu (2021). "Sulforaphane Activates a lysosome-dependent transcriptional program to mitigate oxidative stress." Autophagy 17(4): 872-887.

Medina, D. L., A. Fraldi, V. Bouche, F. Annunziata, G. Mansueto, C. Spampanato, C. Puri, A. Pignata, J. A. Martina, M. Sardiello, M. Palmieri, R. Polishchuk, R. Puertollano and A. Ballabio (2011). "Transcriptional activation of lysosomal exocytosis promotes cellular clearance." Dev Cell 21(3): 421-430.